# Quantifying changes in the T cell receptor repertoire during thymic development

**Francesco Camaglia**[1†], **Arie Ryvkin**[2†], **Erez Greenstein**[2], **Shlomit Reich-Zeliger**[2], **Benny Chain**[3], **Thierry Mora**[1*‡], **Aleksandra M Walczak**[1*‡], **Nir Friedman**[2‡§]

[1]Laboratoire de physique de l'École normale supérieure, CNRS, PSL University, Sorbonne Université, and Université de Paris, Paris, France; [2]Department of Immunology, Weizmann Institute of Science, Rehovot, Israel; [3]Division of Infection and Immunity, University College London, London, United Kingdom

**Abstract** One of the feats of adaptive immunity is its ability to recognize foreign pathogens while sparing the self. During maturation in the thymus, T cells are selected through the binding properties of their antigen-specific T-cell receptor (TCR), through the elimination of both weakly (positive selection) and strongly (negative selection) self-reactive receptors. However, the impact of thymic selection on the TCR repertoire is poorly understood. Here, we use transgenic Nur77-mice expressing a T-cell activation reporter to study the repertoires of thymic T cells at various stages of their development, including cells that do not pass selection. We combine high-throughput repertoire sequencing with statistical inference techniques to characterize the selection of the TCR in these distinct subsets. We find small but significant differences in the TCR repertoire parameters between the maturation stages, which recapitulate known differentiation pathways leading to the CD4[+] and CD8[+] subtypes. These differences can be simulated by simple models of selection acting linearly on the sequence features. We find no evidence of specific sequences or sequence motifs or features that are suppressed by negative selection. These results favour a collective or statistical model for T-cell self non-self discrimination, where negative selection biases the repertoire away from self recognition, rather than ensuring lack of self-reactivity at the single-cell level.

**\*For correspondence:**
thierry.mora@phys.ens.fr (TM);
aleksandra.walczak@phys.ens.fr (AMW)

[†]These authors contributed equally to this work
[‡]These authors also contributed equally to this work

[§]Deceased

## Editor's evaluation

This paper addresses an important question within adaptive immunity, namely whether the T cell receptor (TCR) repertoire of negatively selected thymocytes shares common features. The authors analyze T cell receptor sequences from mice as they progress through positive selection, CD4/CD8 lineage commitment, and negative selection. Thereby they find small but consistent differences between the repertoires at these selection stages, providing arguments that their findings do not indicate any sequence-specific selection.

## Introduction

In order to protect themselves against infection, jawed vertebrates have evolved an adaptive immune system. T lymphocytes play a leading role in this system. Each T lymphocyte expresses a unique T-cell receptor (TCR) capable of binding short protein fragments presented by the host's Major Histocompatibility Complexes (MHC), subsequently triggering clonal expansion and differentiation of immune effector function. The T cell system discriminates pathogen derived 'foreign' proteins from the body's own 'self' proteins, in such a way that an immune response is usually triggered only by peptides from exposure to a potentially harmful threat. We ask if we can identify specific TCR features which allow the system to discriminate foreign and self-peptides.

TCRs are generated in a stochastic assembly process based on random recombinations of genomic templates and additional non-templated insertions and deletions (*Hozumi and Tonegawa, 1976*). The ability to discriminate between self and non-self targets cannot therefore be exclusively inherited, but must at least in part be learned afresh in each individual. This process is widely believed to occur during the development of haemopoetic precursors into mature T cells, which occurs in a specialized microenvironment within the thymus. This process has been studied in considerable detail. T cells precursors first produce a β chain and if the generated chain is functional, the cell proliferates and an α chain is generated. While the TCR chains are being assembled, CD4 and CD8 surface markers are expressed as precursor cells transit to the Double Positive state (DP). DP TCR are subject to thymic selection, a process that tests receptor binding by presenting them with the organism's own proteins, and eliminates very weak binders (positive selection), but also too strongly self-reactive receptors (negative selection) (*Yates, 2014*). During thymic selection, DP cells differentiate into CD4$^+$ or CD8$^+$ cells by keeping expression of only one of these molecules, which determines their function. While this picture is well-established and the maturation trajectory has a well established gene expression signature (*Park et al., 2020*), the TCR sequences removed during thymic selection, which should be manifested as 'holes' in the repertoire, have never been directly observed. The lack of quantifiable signatures of thymic selection, differentiation and proliferation hinders a dynamic description of TCR maturation (*Robert et al., 2021*).

Positive and negative selection imposes upper and lower boundaries on the binding energy of the interaction between TCR and self peptide-MHC complexes (*Kosmrlj et al., 2009*). However, it remains unclear whether every thymocite is exposed to every self-antigen, or how efficient the process of selection is. Negative selection is known to be leaky (*Yu et al., 2015*), letting auto-reactive cells differentiate into regulatory cells (*Bains et al., 2013*; *Wing and Sakaguchi, 2010*). The efficiency of negative selection for the naive conventional (non-regulatory) effector T cell compartment remains unclear (*Yu et al., 2015*; *Gallegos and Bevan, 2006*). Partial or incomplete negative selection may limit its impact on the repertoire.

The difficulty of characterizing selection is partly due to survivor bias when sampling functional immune repertoires in the periphery (*Madi et al., 2017*; *Madi et al., 2014*; *Izraelson et al., 2018*; *Sethna et al., 2017*). To overcome this limitation, we sequenced the TCR repertoire of thymocyte subpopulations isolated from mice carrying a reporter transgene linked to Nur77, a marker of T cell activation both within the thymus and in the periphery. Nur77 expression, in combination with Annexin V, a marker of cell death, allows us to identify cells that are more likely to pass thymic selection, and those that are most likely not to pass selection. Although the CD4+CD8+Annexin V population may still contain some cells which will be negatively selected, but have not yet expressed Annexin V, the overall strategy provides us with a window into the repertoire at various stages of selection. By comparing the sequenced repertoires to statistical models of mouse TCR generation (*Sethna et al., 2017*), and subset-specific models of thymic selection, we searched for specific TCR sequence features that correlate with the different stages of intra-thymic T-cell developement.

## Results

### Tracking T cell development stages by flow cytometry

To identify specific sequence features of TCR during each step of thymic selection, we performed high-throughput sequencing of TCR repertoires from different subpopulations of thymocytes from transgenic Nur77 reporter expressing mice. These mice carry a fluorescent reporter gene which is co-expressed with Nur77, a marker of T cell activation (*Liebmann et al., 2018*). Three genetically identical Nur77 reporter mice were sacrificed at the age of 6 weeks, when thymus development is completed and its cell population is stable (*Gray et al., 2006*). All animals were handled according to Weizmann Institute's Animal Care guidelines, in compliance with national and international regulations. Thymus and spleen were removed, and stained for fluorescence-activated cell sorting (see Materials and Methods). The cells were sorted based on Nur77 reporter expression (to detect activation), Annexin V (to detect early apoptosis) in combination with CD3, CD4, and CD8 cell surface markers. We used the gating strategy illustrated in *Figure 1A, B and C* to isolate double positive DP cells preceding selection (CD4$^+$CD8$^+$, Nur77$^-$, Annexin V$^-$: DP pre), DP cells in the process of being positively selected (CD4$^+$CD8$^+$, Nur77$^+$ Annexin V$^-$: DP pos), DP cells dying by neglect or possibly by

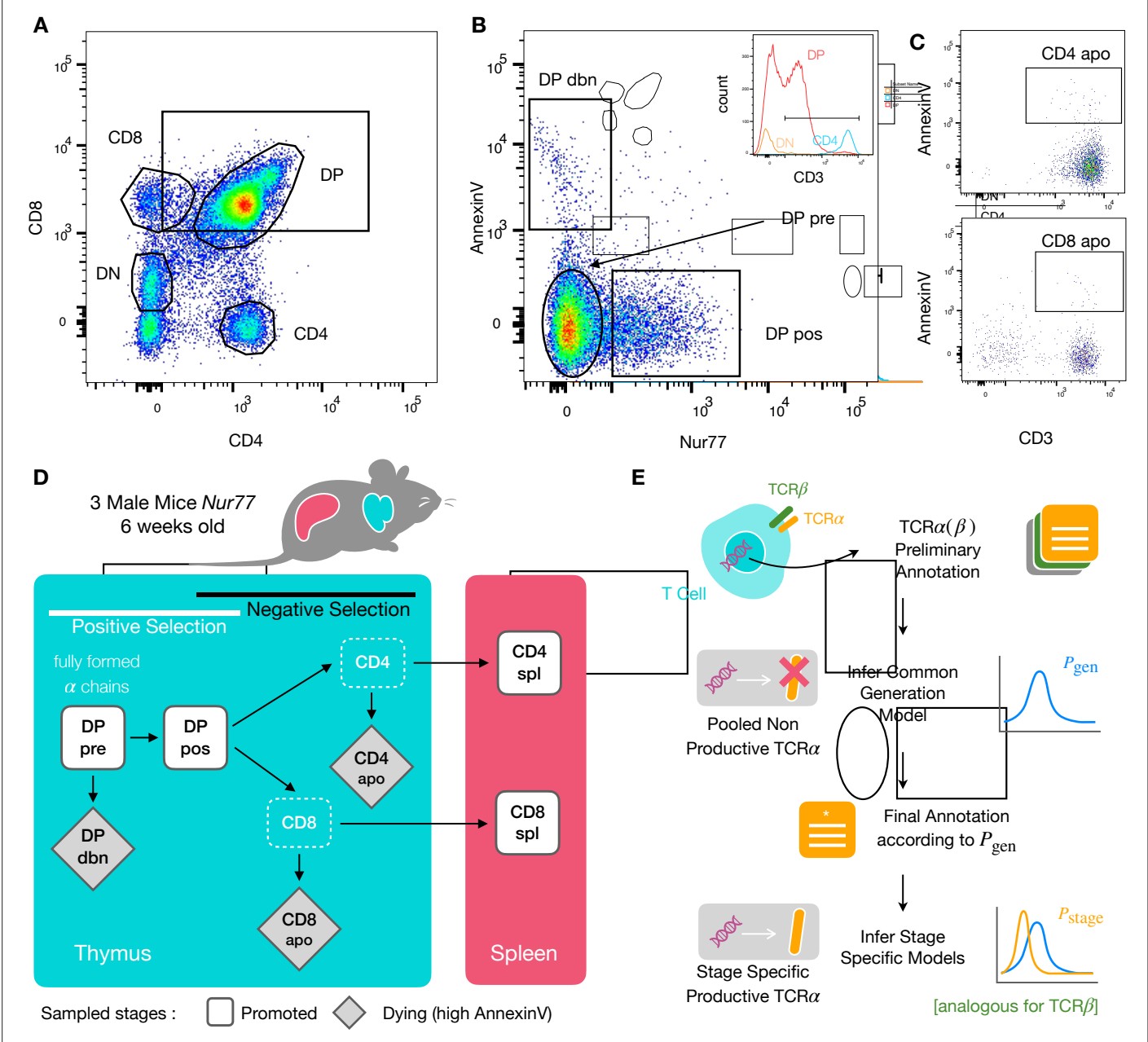

**Figure 1.** Experiment outline and repertoire sampling. (**A**) Flow cytometry scatterplots of T cell population from the thymus according to the markers CD4 and CD8. (**B**) The DP population is separated from DN according to CD3 expression (insert). Cells are then FACS sorted according to the expression of Nur77 and AnnexinV. (**C**) CD4 cells in the spleen (above) and CD8 (below) are FACS sorted according to the expression of CD3 and AnnexinV. (**D**) Schematic evolution of the sampled cell types during thymic maturation. (**E**) Analysis workflow: annotated reads in sampled repertoires are input for model inference (see Materials and Methods). Out-of-frame TCR sequences are pooled from all mice and stages to learn a generation model. In-frame sequences are used to learn maturation stage specific selection models with the generation model as background.

The online version of this article includes the following figure supplement(s) for figure 1:

**Figure supplement 1.** Summary of the RepSeq datasets.

damage during the preparation ($CD4^+CD8^+$, Nur77 Annexin $V^+$: DP dbn); and single positive (SP) cells: $CD4^+CD8^-$, Annexin $V^+$ (CD4 apo), and $CD4^-CD8^+$, Annexin $V^+$ cells (CD8 apo). The Annexin V staining was not very strong and did not give a very clear separation between positive and negative populations. In addition, Annexin $V^+$ subsets may be contaminated by cells that are dying for other reasons than negative selection. Nevertheless, we may still assume that the two apo subsets are enriched in

negatively selected cells. In addition, we sequenced the repertoires of mature (post-selection) single positive SP CD4+ and CD8+ cells from the spleen (CD4 spl and CD8 spl). The proposed differentiation pathway between these populations at different maturation stages are schematically represented in *Figure 1D*. Together, these seven repertoires should contain both the selected thymocytes and the pre-selection repertoires, as well as the thymocytes that fail either positive or negative selection and die in the thymus.

## TCR repertoire sequencing

We sequenced and annotated the TCR repertoires of each subset as described in Materials and Methods. The cDNA of individual α and β genes (TRA and TRB) were barcoded with unique molecular identifiers (UMI) in order to allow for correction of sequencing errors and PCR bias. However, in this analysis we focused on unique sequences (discarding count information) to avoid expression and amplification biases. As a quality control of the whole procedure, we showed that the number of α and β sequences within each population was highly correlated (*Figure 1—figure supplement 1A*). We further verified that the relative fraction of TCRα sequences associated with iNKT cells (identified by TRAV11 and TRAJ18 genes [*Garner et al., 2018*]) is higher in CD4 than in CD8 cells (see *Figure 1— figure supplement 1B*).

We obtained seven datasets for both chains and for each of the three mice. A small fraction of sequences contain stop codons, usually because of a frameshift in the CDR3. These sequences likely come from transcription from a chromosome carrying a nonproductive chain, which is known to persist despite allelic exclusion acting on the TRB locus. The rest of the sequences are assumed to be productive. Since nonproductive TCR owe their survival to the productive gene on the other chromosome, they are not affected by selection. We combined all nonproductive sequences from all subsets to infer a generative mechanistic model of the V(D)J recombination process using IGoR (*Marcou et al., 2018*). Once trained, the model can be used to assign a generation probability $P_{gen}$ to any TCR sequence observed (*Murugan et al., 2012*; *Marcou et al., 2018*; see Materials and Methods and *Figure 1E*).

The datasets contain ~1000–50,000 unique productive sequences per subset (*Figure 1—figure supplement 1C* for the α chain and *Figure 1—figure supplement 1D* for the β chain). Since the 3 mice were isogenic and shared the same MHC haplotype, we expect their repertoires to be subject to the same processes of recombination and selection (*Madi et al., 2014*). Unless specified otherwise, all downstream analyses were therefore carried out on pooled productive TCR sequences from each population from the three individuals to increase statistical power.

## Repertoires from different T cell populations have different statistical parameters

To assess how selection acts at the different maturation stages, we studied the distribution of sequence features in TCRα repertoires. We compared TRAV and TRAJ gene usage at the different maturation stages with each other and with their excepted frequency from the generation model learned from nonproductive sequences, which we will refer to as the pre-selection model or $P_{gen}$. TRAV usage broadly follows the pattern of the pre-selection model (*Figure 2A*), although SP CD4+ repertoires have a lower proportion of TRAV12-2, and most populations have an increased proportion of TRAV7-2. TRAJ gene usage also broadly agrees with the pre-selection model predictions (*Figure 2—figure supplement 2A*), although SP CD8+ repertoires have a lower proportion of TRAJ31, SP CD4+ repertoires have an increased proportion of TRAJ27 and TRAJ32 which is underrepresented in all cell types. For both V and J genes, we see little difference between the repertoires of spleen CD4 and CD8 cells, and their discarded counterparts in the thymus (apo). We also observe strong similarities between all the DP subsets. TRB gene usage follows similar trends, although there are some differences in J gene usage between selected and unselected SP CD4+ and CD8+ cells. Overall, the biases of the recombination process dominate any effects of selection on V and J region usage (*Figure 2—figure supplement 1A*, *Figure 2—figure supplement 2B*).

For both chains, CDR3 amino acid length of SP CD4+ and CD8+ has a sharper distribution compared to earlier maturation stages (DP) (see *Figure 2B*, *Figure 2—figure supplement 1B* and *Figure 2— figure supplement 3*). This has previously been interpreted as a signature of selection due to structural constraints on the pMHC-TCR complex (*Madi et al., 2017*; *Lu et al., 2019*; *Carter et al., 2019*). We also compared the single amino acid usage (excluding the constant regions) across the different

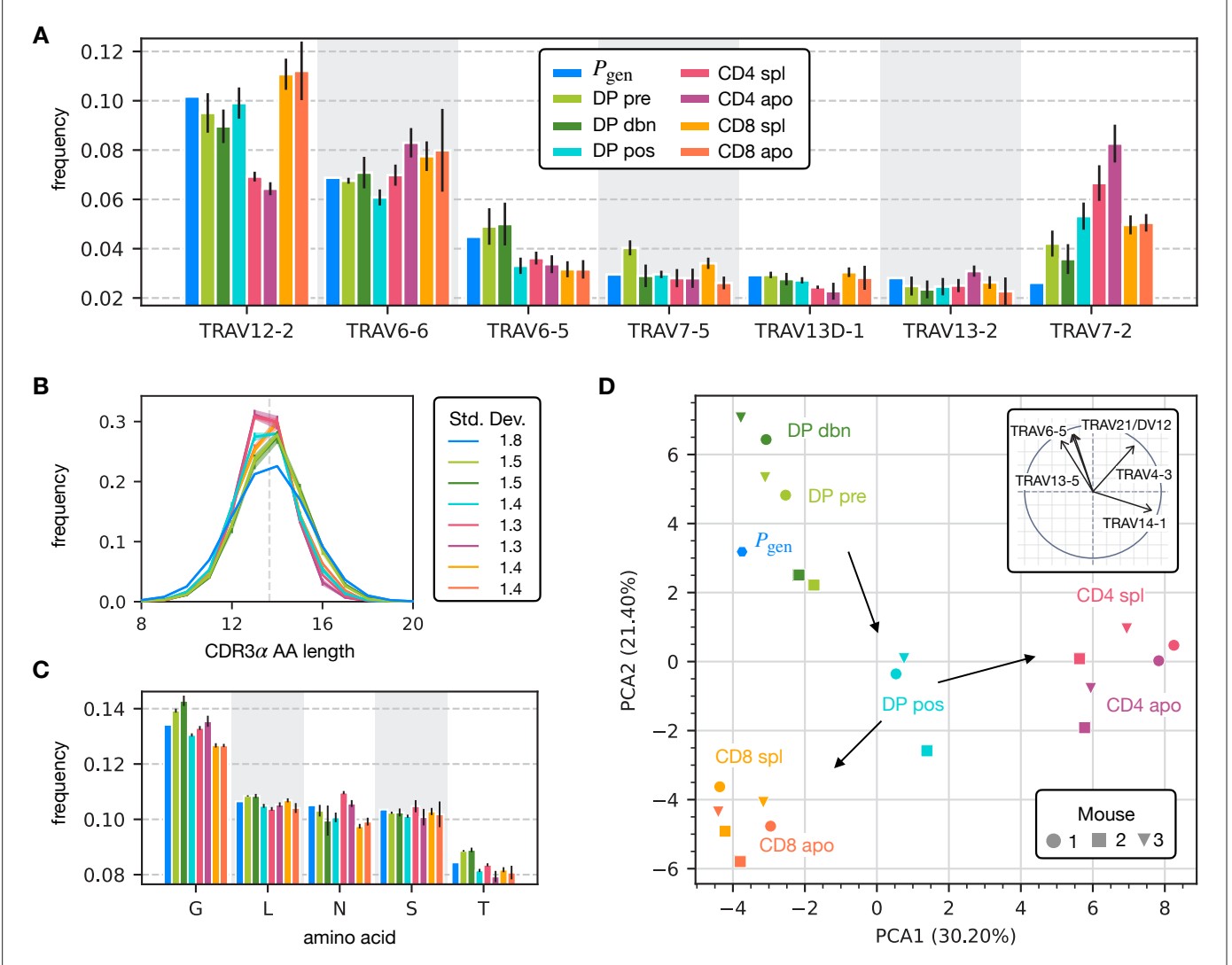

**Figure 2.** Properties of the α chain sequence (the analogous plot for the β chain is showed in *Figure 2—figure supplement 1*). The color code is common to all subplots. (**A**) TRAV gene distribution at different maturation stages compared to the pre-selection model distribution $P_{gen}$ (see *Figure 2—figure supplement 2A* for TRAJ). Only the most frequent according to the $P_{gen}$ model are reported. Errorbars correspond to the empirical standard deviation across the three different mice. (**B**) CDR3 length distribution of TCRα sequences. The errors associated with mouse variability are minor and illustrated via the shaded curves. See *Figure 2—figure supplement 3A* for individual curves. The dashed line is the average CDR3 length from the $P_{gen}$ model. Standard deviations of the average length distributions are shown at right. (**C**) Distribution of the most frequent amino acids at different maturation stages. The counts correspond to the number of observations within the CDR3 (i.e. excluding the first two and the last positions), summed for all the sequences in the subpopulation. Error bars represent the empirical standard deviation across mice. (**D**) Principal component analysis of the TRAV gene distribution at each maturation stage. Insert: projection on the principal axis of the five most abundant TRAV genes (see Materials and Methods). Analogous results for TRAJ are shown in *Figure 2—figure supplement 2C*. Source code available at https://github.com/statbiophys/thymic_development_2022/blob/main/fig2.ipynb.

The online version of this article includes the following figure supplement(s) for figure 2:

**Figure supplement 1.** Analysis of the annotated productive β clonotypes for the different maturation stages.

**Figure supplement 2.** Statistics of the J gene usage and the $P_{gen}$ distributions.

**Figure supplement 3.** Separate amino acid CDR3 length distributions across all stages.

repertoires (*Figure 2C* for α chain, *Figure 2—figure supplement 1C* for β chain). We observe similarities between the DP stages, the CD4 stages and the CD8 stages, as observed for the gene usage. The repertoires from different maturation stages cannot be distinguished by any one individual feature discussed above. However, Principal Component Analysis (PCA) on the TRAV gene usage

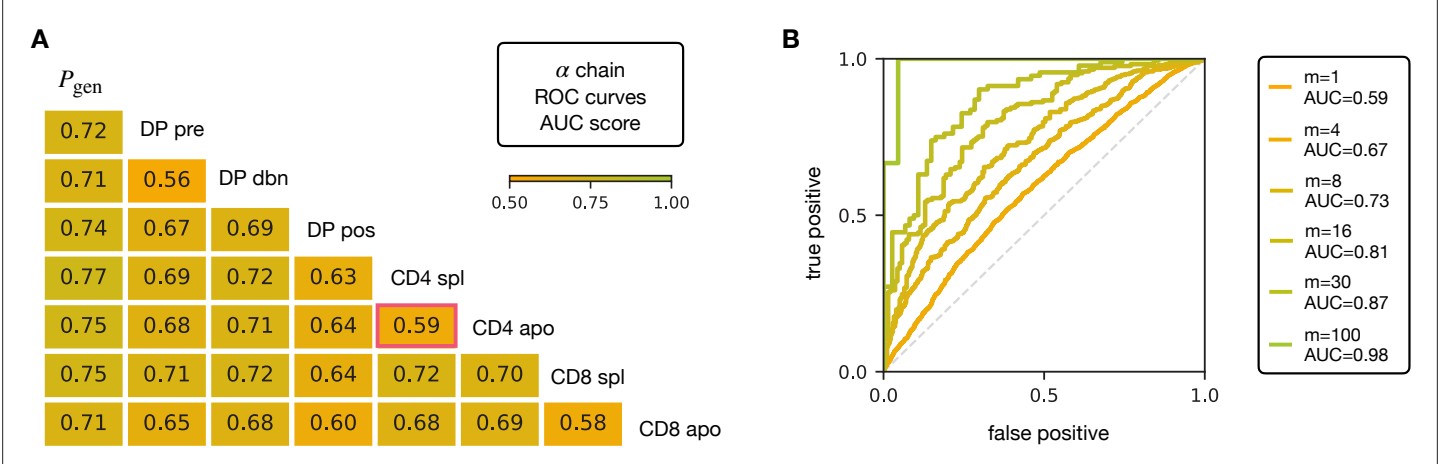

**Figure 3.** AUC scores for the pooled TCRα datasets. (**A**) Area under the curves (AUC) values computed from Receiver Operating Characteristic (ROC) curves of linear classifiers of TCRα between two subsets. The training/testing set is a random subsample containing 70%/30% of the full dataset at a given maturation stage. (**B**) ROC curves for classifying a group of *m* sequences from the same maturation stage, between CD4 spl and CD4 apo (red frame in **A**), illustrating the improvement with increasing number of TCRs. See *Figure 3—figure supplement 1* and (**B**) for the analogous analysis on TCRβ. Source code available at https://github.com/statbiophys/thymic_development_2022/blob/main/fig3.ipynb.

The online version of this article includes the following figure supplement(s) for figure 3:

**Figure supplement 1.** AUC scores for the pooled TCRβ datasets and for an individual mouse.

**Figure supplement 2.** Validation of the stages discrimination.

distributions in individual mice at different stages identified clusters of related cell types (*Figure 2D*). The DP Nur77[-] populations cluster with the pre-selection model, the SP CD4[+] and CD8[+] populations form distinct clusters, and the DP pos Nur77[+] cells, which we hypothesise are cells in the process of positive selection, occupy an intermediate position between these three clusters. This pattern is consistent with the known developmental trajectory as illustrated by the arrows in *Figure 2D*. PCA of TRAJ usage also shows similar clustering patterns (*Figure 2—figure supplement 2C*). The PCA of TRBV and TRBJ usage also discriminates between SP CD4[+] and CD8[+] populations, and from the pre-selection populations, although the overall pattern is less clear (*Figure 2—figure supplement 1D* and *Figure 2—figure supplement 2D*). The overall distribution of TCR generation probabilities, $P_{gen}$, does not change from the pre-selection and post-selection thymic stages to the mature peripheral SP repertories (*Figure 2—figure supplement 2E, F*), consistent with previous reports comparing thymic and peripheral repertoires (*Sethna et al., 2017*). In summary, the effects of selection impose subtle changes on the pattern of TCR variable gene usage, which cannot be adequately captured by looking at any single V or J gene, but only by a combination of features.

V and J gene usage, and CDR3 length are coarse grained measures of a TCR repertoire. We therefore explored whether the repertoires of different maturation stages could be linked to more precise features of the TCR sequence, in particular incorporating the sequence of the CDR3. We encoded each TCR as a sparse {0,1} binary vector $\vec{\sigma}$ which captures V gene, J gene and CDR3 amino acid sequence (for details see Materials and Methods). We then trained a logistic regression model on the set of $\vec{\sigma}$ from repertoires of different subsets. We trained and tested the classifier to distinguish pairs of repertoires from different subsets. The classifier achieved only moderate Area Under the Curve (AUC) of the Receiver Operating Characteristic (ROC) scores (*Figure 3A* for TCRα, and *Figure 3—figure supplement 1A* for TCRβ), in agreement with previous studies (*Emerson et al., 2013*; *Isacchini et al., 2021*). We verify that this result is not an artifact introduced by pooling repertoires of different mice, by testing the same techniques on the individual with the largest datasets (mouse 3). The AUC scores for the α and β repertoires are shown respectively in *Figure 3—figure supplement 1C, D*.

Controls in which population labels were shuffled, resulted in AUC close to 0.5 (*Figure 3—figure supplement 2A, B* for the α chain, *Figure 3—figure supplement 2C, D* for β). The results shown in *Figure 3* indicate that the TCR populations differ at a statistical level (i.e. have different distributions of sequence features), but that each individual TCR is only a weak predictor of repertoire class. However,

better classification efficiencies can be achieved by combining the predictions from sets of TCRs. For example, multiplying the predictions from 30 TCR sequences from the same repertoire (*Figure 3B*), we can distinguish CD4 spl and CD4 apo TCRα with an AUC score of >0.85; see *Figure 3—figure supplement 1B* for TCRβ. Thus, statistical properties of a repertoire can distinguish it from another repertoire, even when the feature distributions of individual TCRs are largely overlapping.

## Selection models and *n*-grams capture the relations between the stages of thymic development

A number of studies have highlighted the importance of short amino acid motifs (*k*-mers or *n*-grams) within the CDR3 sequence in determining TCR specificity (*Thomas et al., 2014*; *Sun et al., 2017*; *Cinelli et al., 2017*) (see *Figure 4A*). Specifically, *n*-grams can be used to reduce the dimensionality of the TCR space, while capturing amino acid correlations or patterns which might play a role in antigen recognition. We therefore counted the frequency of *n*-grams in each repertoire. We excluded from the analysis the most conserved regions (the first two positions in the CDR3 that are usually a cysteine and alanine, and the last one, typically a phenylalanine). We then used these *n*-gram frequency distributions to calculate the diversity of the repertoire as quantified by the Shannon entropy $S$ (see Materials and Methods). In practice, the Shannon entropy is computationally too expensive to calculate exactly for very large data sets, and we therefore restricted our analysis to *n*-grams of length 4 or less, using the approximate Nemenman-Shafee-Bialek (NSB) entropy estimator (*Nemenman et al., 2002*) to correct for finite sampling bias (see Materials and Methods). This estimator outcompetes alternative entropy estimators on synthetic data (*Figure 4—figure supplement 1*). Our analysis combines together CDR3 of different amino acid lengths which may influence the entropy measurements. However, detailed analysis of the entropy of DP repertoires, using different CDR3 lengths separately, demonstrated that the differences observed due to the lengths effects were small compared to error due to sequencing (*Figure 4—figure supplement 1*). Another advantage of the Nemenman-Shafee-Bialek estimator is that it was shown to converge at the sizes of the smallest datasets ($\sim 10^3$-$10^4$ clonotypes), as reported in *Figure 4—figure supplement 2*. Once computed the set of entropy measurements based on *n*-gram frequencies for each different repertoire, we compared the data-derived entropy measurements with the prediction of a simple generative model of each repertoire which treated each feature of each TCR (V gene, J gene and each CDR3 amino acid) as independent. Taking the set of TCR vectors $\vec{\sigma}$ we fitted a set of parameters $E_{\text{stage}}$ by maximising the posterior probability over all of the TCRs for each repertoire separately $P_{\text{stage}}(\vec{\sigma}) = (1/Z)e^{-E_{\text{stage}}(\vec{\sigma})}P_{\text{gen}}(\vec{\sigma})$, where $P_{\text{gen}}(\vec{\sigma})$ are the pre-selection generative probabilities for all the TCRs, $E_{\text{stage}}(\vec{\sigma})$ is a linear function of the features (*Elhanati et al., 2014*; *Sethna et al., 2020*), and $Z$ is a normalization factor (*Figure 1E* and Materials and Methods). The enrichment factors $E_{\text{stage}}(\vec{\sigma})$ encode the intuition that due to selection, a given TCR in a given repertoire is seen with higher or lower frequency than expected by the pre-selection generation model. Once we had learnt the enrichment factors for each repertoire, we used the resulting model to generate in silico synthetic repertoires of $3 \times 10^6$ TCRs, and recalculated *n*-gram frequency distributions and entropy estimates for each synthetic repertoire.

The comparison of the estimated entropy for each *n*-gram length, and each subpopulation of T cells, using both directly data-derived and model-derived repertoires is illustrated for TCRα (*Figure 4B*) and TCRβ (*Figure 4—figure supplement 3A*) chains. An upper bound for the entropy is given by uniformly distributed amino acids, $S_{\text{max}}/n = \log_2 20 \sim 4.3$ bits, while amino acids distributed according to their frequency in the overall vertebrate proteome gives a slightly smaller value of ~4.2 bits per position (*King and Jukes, 1969*). Both the observed and model-derived entropies are less than this maximum even for single amino acids (*n*-grams of length $n = 1$), and decrease further with *n*-gram length (see *Figure 4—figure supplement 3B, C*). This reflects strong bias on the abundance of individual amino acids, and strong correlations between amino acids within the CDR3 which are observed in all CDR3 repertoires, and are captured by the frequency distribution of the longer *n*-grams. Two additional important points can be observed. First, the entropy of the repertoires after selection and lineage commitment (in the single positive populations) is less than the earlier pre-selection DP repertoires, which match closely the entropy of the pre-selection generative model (shown by the dotted line for each *n*-gram length). This decrease becomes more evident with longer *n*-gram length (the circles lie below the dotted lines). Thus, as predicted, selection does impose some decrease in repertoire diversity, although this is a much smaller effect than the decrease in diversity imposed by

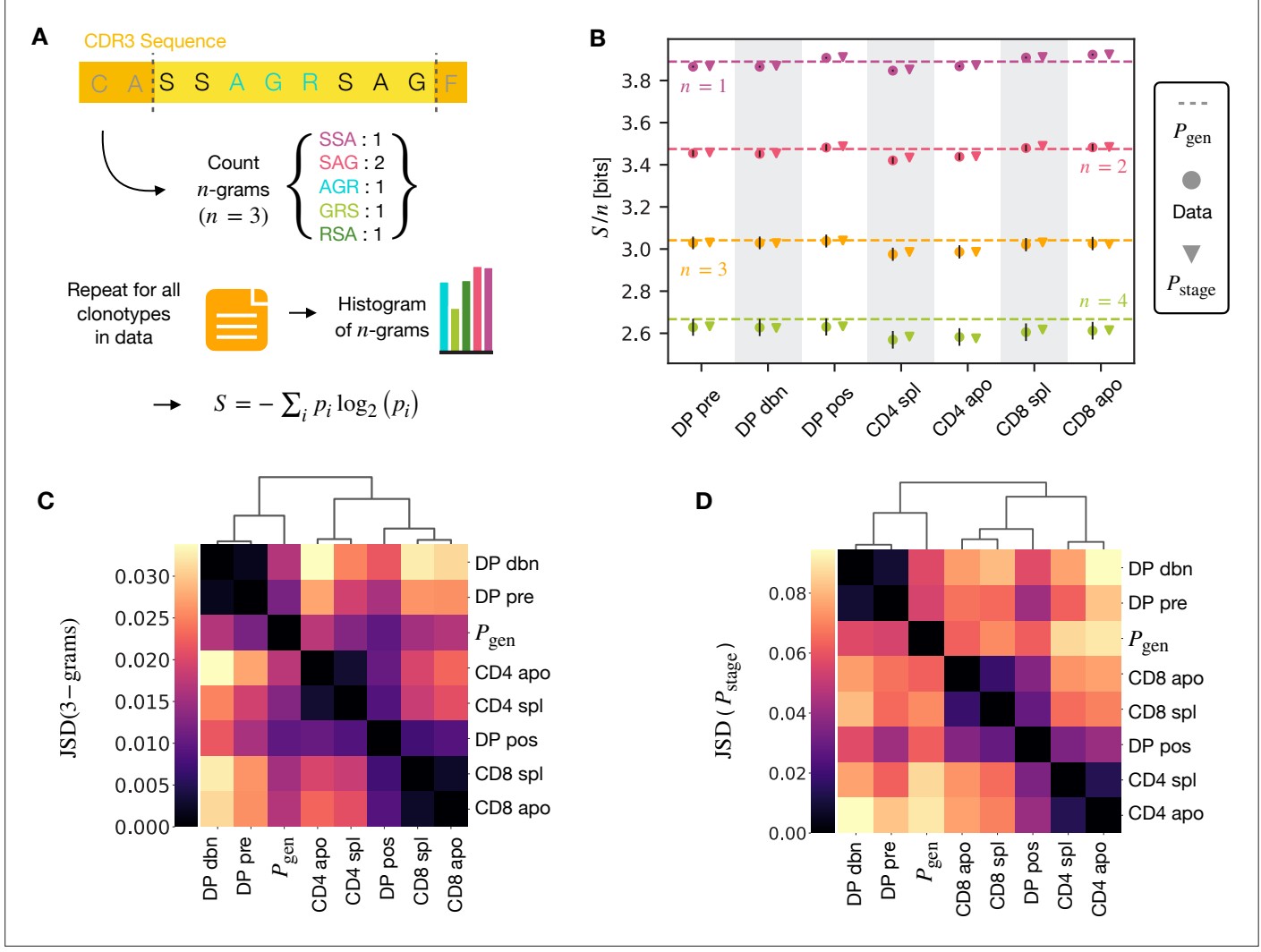

**Figure 4.** *n*-gram frequency discriminates between repertoires. (**A**) *n*-gram definition. We count how many times *n*-gram amino acid subsequences are seen in the CDR3 across a repertoire. (**B**) Shannon entropy $S$ of the *n*-gram distributions normalized by *n* for the maturation stages. The entropy is estimated with the *Nemenman et al., 2002* estimator and it is expressed in bits. The error on the estimated Shannon entropy from data is estimated from the sequencing error (see Materials and Methods). (**C**) Clustering according to Jensen-Shannon divergence between the 3-gram distributions computed from the selection model $P_{\text{stage}}$ on synthetic repertoires. Dendrogram are computed with the Ward method (see Materials and Methods). (**D**) Clustering based on Jensen-Shannon divergence for the full $P_{\text{stage}}$ selection model using $P_{\text{stage}}$. Source code available at https://github.com/statbiophys/thymic_development_2022/blob/main/fig4.ipynb.

The online version of this article includes the following figure supplement(s) for figure 4:

**Figure supplement 1.** Comparison of different entropy estimators and of the dependence on the CDR3aa length choice.

**Figure supplement 2.** Convergence of the *n*-gram entropy estimations.

**Figure supplement 3.** Shannon entropy on β-grams and entropy dependency on *n*.

**Figure supplement 4.** Jensen-Shannon divergence between *n*-gram distributions.

**Figure supplement 5.** AUC values computed from the ROC curves of the linear classifiers learnt over n-grams features.

**Figure supplement 6.** Measure of the Shannon entropy using the full stage models.

**Figure supplement 7.** Logo plots for the relative enrichment of positional amino acid usage.

**Figure supplement 8.** Hydrophobic score at different stages and AUC scores of classifiers on hydrophobic features.

the generation process itself. The second key observation is that the entropy calculated directly from $n$-gram frequency in the data is very similar to that of the synthetic repertoires generated using the linear generative models in which individual TCR amino acids are treated as independent variables. Thus, at least at the level of diversity of $n$-grams, there is no evidence that selection at any step involves complex sequence motifs, or amino acid interactions, which would not be captured by the linear model. We looked in more detail at the $n$-gram ($n = 3$) distributions derived by the linear selection models for the different maturation stages. A plot of the Jensen-Shannon divergences (JSD) between all pairwise comparisons largely recapitulated the expected relationships between the subsets, with DP pre and DP dbn clustering with the pre-selection generative model, while the single positive CD4 and CD8 populations clustered separately, and DP pos have an intermediate position (*Figure 4C*). A comparison for both TCRα and β for different $n$ is shown in *Figure 4—figure supplement 4*. Since some differences between populations were seen even for amino acid usage ($n = 1$), we compared the discriminatory power of models based on $n$-grams with $n = 1$ and $n = 3$ (*Figure 4—figure supplement 5*). The 3-gram model outperformed the 1-gram model in almost all cases. We can go beyond $n$-grams and use the subset-specific $P_{stage}$ models to predict the entropy of the full sequence (Materials and Methods), shown in *Figure 4—figure supplement 6A* for $\alpha$ and *Figure 4—figure supplement 6C* for β. This entropy is substantially reduced from generation to the DP stages, and further reduced in the single positive stages, especially in CD4$^+$ subsets. We also computed the JSD of the distributions $P_{stage}$ between subtypes (*Figure 4C* for TCRα and *Figure 4—figure supplement 6D* for TCRβ). These JSD showed similar patterns as with $n$-grams, except for CD8$^+$ spleen cells showing more similarity to the $P_{gen}$ distribution in TCRβ. Note that the absolute values of the entropies and JSD are larger, since they include information about longer sequences, with additional V and J gene usage information. In summary, we fitted the data with a set of stage-specific generative models based on linear weighted combinations of TCR sequence features. The repertoires generated by this model accurately estimate the sequence and $n$-gram entropy derived directly from the data, and generate repertoires which differ in a small but reproducible manner from each other. The magnitude of these differences reflect the expected developmental relationships between the different populations.

## Models capture modulations of hydrophobic residues in different subpopulations

We inspected single amino acid usage in terms of the model marginals to check for relative positional enrichments between pairs of repertoires (*Equation 3*), but we did not observe any striking signal for amino acid charge properties. The logo plots with a visualization of this results are shown in *Figure 4—figure supplement 7*. Hydrophobic residues in the central positions of the CDR3 have been reported to be enriched in the TCRs of regulatory versus conventional T cells (*Lagattuta et al., 2022*). This suggests hydrophobicity may function as a proxy for auto-reactivity, and might be enriched in cells selected for negative selection (*Stadinski et al., 2016*; *Daley et al., 2019*). To test this idea, we defined a stage-specific hydrophobicity score $U$, obtained by summing the enrichment factors of hydrophobic residues CFILMWY at central positions of the CDR3 as learnt by our model at each stage (see *Equation 4* in Materials and Methods).

We observe a clear increase of this score from DP pre to DP pos, suggesting that positive selection introduces a bias toward more hydrophobic TCRs (*Figure 4—figure supplement 7* and B). The score also decreases in the single positive sets (CD4 and CD8), in agreement with the known role of negative selection to prune too strongly self-reactive T cells (*Butler et al., 2013*). Finally, AnnexinV+ single positive sets ('apo') show a slightly higher score than their respective spleen ('spl') sets (with the exception of the CD8 α chain scores). Overall, these changes in hydrophobicity are consistent with the hypothesised position of the different populations defined in our study in the stages of TCR selection.

Note that this score (like other scores found in the literature *Isacchini et al., 2021*; *Lagattuta et al., 2022*) is statistical and can not be used to classify individual sequences. To assess how much of single-sequence discriminability is explained by the presence of hydrophobic residues, we then introduced an empirical 'hydrophobicity index' $u$, here defined as the number of hydrophobic residues (again CFILMWY) contained in the CDR3, normalized by its amino acid length (see Materials and Methods, *Equation 5*). The classifiers using this feature yielded poor performance (*Figure 4—figure supplement 8C, D*), worse than the 1-gram models (*Figure 4—figure supplement 5A, B*).

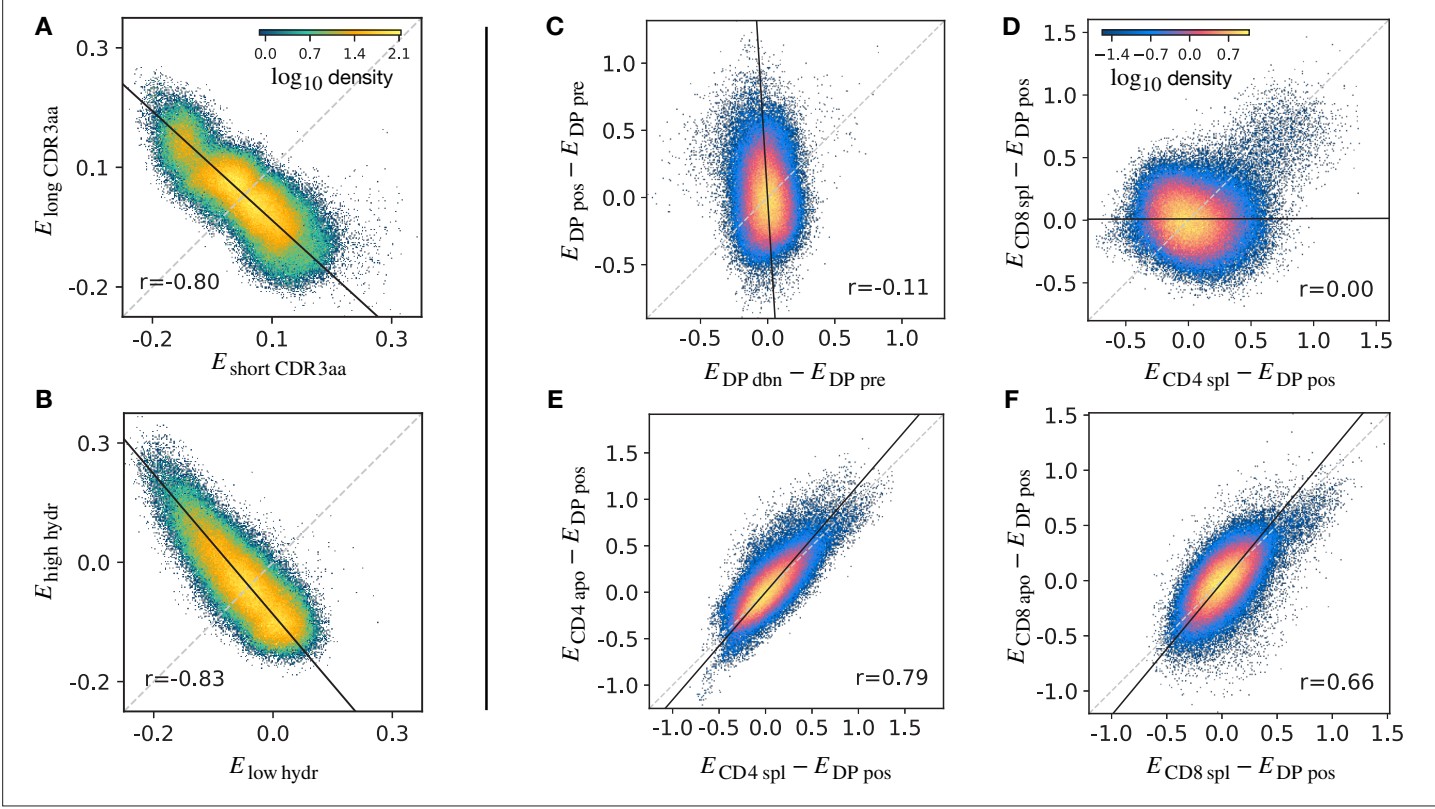

**Figure 5.** Density scatter-plots of TCRα sequences comparing the selection energies learnt at two different stages. (**A**) Synthetic example of soft discrimination between 'short' and 'long' CDR3, where sequences are randomly assigned into either of the two populations with a bias that depends on their CDR3 length. The density scatter plot shows a clear anti-correlation between the selection energies learnt from these two populations. Yet, sequence classification is imprecise, as quantified by the low AUC = 0.57. The parameters chosen for the filter in this example are $L_0 = 13$ and $h = 2$. (**B**) Synthetic example of soft discrimination between 'low' and 'high hydrophobic' CDR3 showing clear anti-correlation between these two populations. Sequence classification is again poor AUC = 0.60. The parameters chosen for the filter on the 'hydrophobic index' $u$ in this example are $u_0 \simeq 0.2$ (the median value over a set of $P_{gen}$-distributed sequences) and $h = 1$. (**C**) The differential enrichment parameter of each TCR calculated according to $P_{DP\,dbn}$ model is plotted against the energy calculated against the $P_{DP\,pos}$ model. To correct for bias imposed by the TCRα generation process, the DP pre energy, which encodes background selection common to both stages, is subtracted. The black line is the direction of the major eigenvector of the dots moments matrix. The value $r$ reported in each plot is the Pearson's correlation coefficient (see Materials and Methods). (**D**) Differential enrichment parameter according to CD4 spl and CD8 spl models, relative to DP pos. (**E**) Differential enrichment parameter according to CD4 spl and CD4 apo models, relative to DP pos. (**F**) Differential enrichment parameter according to CD8 spl and CD8 apo models, relative to DP pos. Source code available at https://github.com/statbiophys/thymic_development_2022/blob/main/fig5.ipynb.

The online version of this article includes the following figure supplement(s) for figure 5:

**Figure supplement 1.** Differential increments scatterplots for all pairs of stages.

## Discriminatory power of thymic selection

The stage-specific enrichment factors in the generative models described above can be considered as capturing the combination of features which drives a particular selection step. A prediction of this idea is that, at each selection point, the TCRs which are selected and those which are not would have a distribution of model probabilities ($P_{stage}$) which are anti-correlated. For example, a TCR that is present in the DP pos repertoire but "forbidden" from the CD4 repertoire (e.g. because of cross-reactivity to a Class II self pMHC) would be expected to have a large positive $P_{DP\,pos}$ and a $P_{CD4\,spl} \approx 0$, reflecting the large enrichment factor between these two populations. A toy example illustrating this idea is illustrated in (**Figure 5A**). We consider a simple model in which TCRs are selected according to their CDR3 length into a 'long' population with probability $P(long|L) = L^h/(L^h + L_0^h)$ and into a 'short' population otherwise. We apply this selection process in silico to $P_{gen}$-generated TCRs, and fit a separate $P_{population}$ model on the synthetic sequences found in each subset. We then calculate $E_{population}$ for each TCR according to both subset models, and plot these values against each other. The distribution

of enrichment strengths according to the two models are clearly anti-correlated (*Figure 5A*). In other words, if a TCR is more likely to be classified as a 'long' sequence, it is in general less likely to be classified as a 'short' one. Interestingly, however, the enrichment strengths distributions from the two models are significantly overlapping. As a result, attempts to classify individual TCRs according to their enrichment strengths is poor, AUC~ 0.57. We then consider a different toy model where TCRs are selected according to the empirical hydrophobicity index $u$ (*Equation 5*). Similarly, we choose to generate a synthetic 'high hydrophobicity' (HH) population filtering $P_{gen}$-generated TCRs with a probability $P(HH|u) = u^h/(u^h + u_0^h)$, otherwise 'low hydrophobicity'. Repeating the analysis performed with the length example, we observe in the corresponding scatterplot that enrichment strengths are again anti-correlated (*Figure 5B*).

We extended this approach to look for relationships between enrichment strengths for TCRs at different developmental stages. Since all cells pass through a preceding selection stage, we must consider it as a common background distribution for all the successive thymic stages. We therefore considered the differential enrichment parameter $E_{stage} - E_{pre-stage}$, a linear operator which predicts whether a sequence is more or less likely to be present in a particular developmental stage as compared to the previous stage. We generated a set of sequences using the generation model $P_{gen}$ (thus with no selection bias), and then computed differential enrichment parameters for each TCR according to all the stage specific models. The full set of pairwise correlations between enrichment values for the different populations relative to $P_{DP\ pre}$ are shown in (*Figure 5—figure supplement 1A* for TCRα and *Figure 5—figure supplement 1C* for TCRβ). The DP dbn repertoire showed a narrow distribution of values, which was uncorrelated to any other subset, in particular to DP pos (*Figure 5C*). This would be consistent with the DP dbn repertoire containing a random sample of the DP pre repertoire, unrelated to its TCR sequence. To check if the signal coming from DP pos stage is the principal cause of the high correlation between the single positive stages, we repeated the analysis for CD4 and CD8 using $P_{DP\ pos}$ as the common background distribution (the full set of scatterplots for TCRα is shown in *Figure 5—figure supplement 1B*, in *Figure 5—figure supplement 1D* for TCRβ). There was therefore no evidence of selection pressure operating on TCR sequence to distinguish these two populations. The correlation between the CD4$^+$ and CD8$^+$ subsets was negligible ($r \sim 0$), suggesting that the selection pressures operating on the two populations are distinct (*Figure 5D*). The spleen SP and the thymic apo populations were also highly correlated for both CD4$^+$ and CD8$^+$ cells ($r = 0.79$ for CD4 spl vs CD4 apo, in *Figure 5E*; $r = 0.66$ for CD8 spl vs CD8 apo, in *Figure 5F*). Similar results are obtained for the sequences of the β chain (*Figure 5—figure supplement 1C D*). In contrast to the examples illustrated above, most plots showed a positive correlation between enrichment values for two models. Thus a common dominant selection process is driving the repertoire shift between the DP pos and all subsequent stages, which dominates the impact of individual stage-specific selection processes. In summary, the TCR enrichment value distributions differ between different thymic populations, but do not show evidence of dominant exclusive sequence-based selection operating at any step of the selection process.

## Discussion

Thymic selection is often portrayed as a simple discrimination process that eliminates TCRs capable of strongly binding any self-peptide, while promoting TCRs that bind them weakly. However, this simple picture has been challenged and the fidelity of the negative selection process and the proportion of the self-repertoire which can effectively be scanned by each individual thymocyte during the window of negative selection remains incompletely understood (*Yu et al., 2015*; *Gallegos and Bevan, 2006*). If significant number of T cells escape negative selection and enter the peripheral repertoire, no sequence feature will unambiguously distinguish TCRs from pre and post-selection repertoires. Many efforts have been made to connect TCR sequences to peptide recognition (*Weber et al., 2021*; *Montemurro et al., 2021*; *Isacchini et al., 2021*). However, these approaches cannot yet be used to define the target peptidome of entire repertoires. Here we take the complementary approach, by looking for TCR sequence features that are linked to thymic selection.

Although there has been a lot of work on understanding and modeling thymic development (*Yates, 2014*; *Robert et al., 2021*) our study presents the first comprehensive analysis of TCR repertoire of different developmental stages of thymic maturation. By incorporating a reporter for the activation marker Nur77, which is activated during thymic selection, and an early marker of apoptosis, Annexin

V, we were able to enrich for identifying subpopulations during the process of positive or negative selection. Although this more sophisticated strategy in principal allows the unbiased isolation of the major stages of thymic selection, some limitations remain. For example, the time interval during which negatively selected cells survive after they received their instruction to go into apoptosis may depend on signal strength. If strong TCR singal strength translates into short subsequent lifetime, then the AnnexinV+ cells sorted may be enriched for cells receiving a rather weaker negative signal. We examined the repertoires from two perspectives. In the first part of the paper, we compare statistical properties of the sequences of the repertoires using features of different dimensionalities, which include V gene, J gene and CDR3 length frequency distributions, and individual CDR3 sequences represented as sparse {0,1} binary vectors. The analysis incorporated both coarse-grained (V, J and CDR3 length) and fine-grained (individual CDR3 sequence) features, and the results were remarkably consistent. No single feature adequately discriminated between any pair of repertoires. Combination of features when averaged across a repertoire did show subtle but reproducible differences between repertoires, which could be used to discriminate between subpopulations using both unsupervised (PCA) and supervised (logistic regression) analysis. Furthermore, the difference between these statistical parameters captured the known developmental trajectory of thymic development, illustrated schematically in *Figure 1D*. Interestingly, the smallest distances observed were between mature CD4 or CD8 cells, and their thymic SP negatively selected (apo) counterparts. This suggests that negative selection of single positives is only weakly associated with the sequence properties of single TCRs, or at least single chains. It is in principle possible that larger differences exist in the paired α-β repertoires, which would not be detectable in either the alpha or beta repertoires alone, but previous work on the functional alpha-beta repertoire has suggested that pairing was largely random, with weak associations between some germline genes (*Grigaityte et al., 2017*; *Dupic et al., 2019*).

An additional possibility which must be considered is that Annexin V staining does not exclusively capture the negatively selected population, but also identifies cells which were damaged during the preparation. Contamination of the AnnexinV+ population by these damaged or dying cells will weaken the selection signature observed, although the fact we do manage to discriminate between the apo and spleen subpopulations (*Figure 3A*) indicates that these differences, however small, do exist. Conversely, cells marked for deletion may not have the time to express Annexin V, so that the DPpos subset may contain cells that are being negatively selected against, in addition to cells that are being positively selected (*Stritesky et al., 2013*).

A limitation of our sorting strategy is that we do not identify Treg from conventional CD4[+] T cells. It has been suggested that regulatory T cells (Tregs), which are more auto-reactive and should thus bear the same marks as the cells that fail negative selection, have distinctive TCR features, notably the presence of hydrophobic residues at key positions (*Stadinski et al., 2016*; *Daley et al., 2019*). TCR scores based on more detailed features than hydrophobicity have been proposed (*Isacchini et al., 2021*; *Lagattuta et al., 2022*). We note that these scores are statistical and do not classify individual sequences. Consistent with these previous results, we can project our model parameters to build a single hydrophobicity index, which we observe to be significantly increased in positively selected cells (DP pos) versus DP pre, and decreased in single positive sets (*Figure 4—figure supplement 8A, B*). Beyond hydrophobicity, it remains an open question whether the features that drive Treg fate are the same that drive negative selection.

Although the statistical properties of the repertoires differed between subpopulations, it was not possible to classify individual TCRs at high accuracy. As discussed above, this may in part be due to the fact that the populations we define only imperfectly correlate with their fate and self-reactivity. However, the differences between CD4[+] and CD8[+]repertoires, which are much less likely to be affected by issues of functional or physical cross-contamination, are also seen only at a statistical population level, and not an individual TCR sequence level. Learning the collective properties of at least a few dozen TCRs was required in order to achieve good discrimination between repertoires.

The statistical population-level differences between populations of thymocytes and mature T cells which we observe is reminiscent of previous models emphasising the importance of collective, rather than individual T cell behavior. *Butler et al., 2013* proposed that a minimum number of T cells must collectively recognize a peptide to trigger a response, proposing quorum sensing as a mechanistic explanation of this collective decision making. Recent experiments have confirmed that quorum sensing between TCRs can occur, mediated via cytokine signaling (*Polonsky et al., 2018*),

and estimating a minimum quorum size of activated T cells to be 30. Our results suggest that thymic selection imposes only a rather weak selective pressure on the repertoire, which is consistent with *Butler et al., 2013*'s hypothesis that most self-peptides are not screened by TCR during negative selection. Our results are consistent with their model, in which even a subtle depletion, rather than complete elimination of non-self TCRs, may still translate into robust self/non-self discrimination in populations of reactive TCRs. Self versus foreign peptide discrimination by TCR is somewhat the conjugate task of self-reactive versus a non self-reactive T cell discrimination during negative selection. While the performance of the two tasks cannot be directly compared at first sight, they are related in that both are impaired by a factor $(1 - f)$ due to partial screening of self peptides, where $f$ is the fraction of self-peptides that are presented during thymic developement. The common point is that even when $f$ is small, the law of large numbers can rescue the discrimination task when there are multiple observations. In Materials and Methods, we argue using the model of *Butler et al., 2013* how the idealized performance of repertoire discrimination using multiple ($m$) TCR (akin to the task of *Figure 3B*) may be compared to the task of telling self from foreign peptides in the periphery, when the number of T cells specific to one particular peptide and recruited to the site of infection is $m \times \bar{n}$, where $\bar{n}$ is the average number of self-peptides recognized by a random TCR. While those numbers cannot be applied directly to the results of *Figure 3B*, which are based on an imperfect classifier from a single chain, they give a sense of how the same principle of discrimination apply to both cases.

In the second part of the study we explore in more detail whether we can discover any evidence that thymic selection depends on specific sequence motifs (i.e. a strong correlative structure between CDR3 amino acids). For this purpose, we build on our previous work which have established a framework for the development of generative statistical models of repertoire generation, based firmly on a mechanistic understanding of TCR generation and selection. Specifically, we construct models which incorporate only linear combinations of CDR3 sequences to capture the selective process which can transform one repertoire into another. These models produce an 'enrichment factor' for each TCR which estimates its relative likelihood of being in a particular stage-specific population. Intuitively, one can consider these factors as capturing the probable enrichment or depletion of a TCR with a particular sequence when comparing two repertoires. We demonstrate that these linear models effectively capture the progressive decrease in repertoire diversity which we observe in the preselected DP to the SP transition. They also effectively capture the known developmental relationships between the thymic subpopulations. Thus we find no evidence that complex non-linear amino acid sequence interactions are required to explain the observed changes in repertoire observed in our data. We also compared the distributions of enrichment factors between populations. We demonstrate that, contrary to the predictions of a strong binary selection model, we do not observe any negative correlation between enrichment factor distributions between selected and non-selected repertoires. Instead, we observe a set of positive correlations, revealing a dominant conserved selection process spanning the developmental stages between pre-selection DP and mature SP. Consistent with the clustering data discussed above, we find strong correlation between the enrichment factor distributions of mature SP and thymic negatively selected population, and no evidence of binary selection between these two populations.

In conclusion, we report a comprehensive analysis of the TCR repertoire at various stages of thymic development. We then combine data-driven and model-based analysis of these repertoires. Our conclusions are incompatible with a model of thymic developments which involves a sequence of clear-cut binary selection processes, based on TCR sequence features. Rather, our data suggest a probabilistic fuzzy decision making process at each selection step. We propose that this model is compatible with robust self/non-self discrimination, if T cell responses to antigen are governed by collective quorum based decision making. Further experimental and theoretical work is required to test these hypotheses, which have fundamental implications for strategies to modulate the immune response for prophylaxis or therapy of human disease.

## Materials and methods
### Animals
The experiment was carried out using three 6-week-old male inbred Nur77-GFP/Foxp3-mCherry (C57BL/6 background) (*Moran et al., 2011*). The cross was bred and maintained at the Weizmann

**Table 1.** Cell sorting based on fluorochrome-labeled mouse antibodies.

| Sample\Marker | CD4 | CD8 | CD3 | AnnexinV | Nur77 |
|---|---|---|---|---|---|
| DP pre | + | + | + | - | - |
| DP pos | + | + | + | - | + |
| DP dbn | + | + | + | + | - |
| CD4 apo | + | - | + | + | |
| CD8 apo | - | + | + | + | |

institute. This study was performed in strict accordance with the recommendations in the Guide for the Care and Use of Laboratory Animals of the National Institutes of Health. All of the animals were handled according to approved institutional animal care and use committee (IACUC) protocols (#21661115–2) of the Weizmann Institute of Science. The protocol was approved by the Committee on the Ethics of Animal Experiments of the Weizmann Institute of Science. Every effort was made to minimize suffering.

## Sample preparation and T cell isolation

Thymocytes and splenocytes were isolated from Nur77-GFP/Foxp3-mCherry 6-week-old mice. Erythrocytes were removed by hypotonic lysis in ammonium chloride. Thymocytes were stained with fluorescent antibodies, and sorted using a flow cytometer as described below. Splenic CD4 and CD8 cells were purified in two steps: (1) CD4+ positive selection (CD4 (L3T4) MicroBeads, mouse, # 130-117-043, Miltenyi) to generate the 'CD4 spl' samples (2) the negative cells fraction were further selected for the CD8+ positive cells (CD8a (Ly-2) MicroBeads, mouse, # 130-117-044, Miltenyi Biotec) to generate 'CD8 spl' samples.

## Flow cytometry analysis and cells sorting

The following fluorochrome-labeled mouse antibodies were used according to the manufacturers' protocols: PerCP/Cy5.5 anti-CD4, PB anti-CD8, PE/cy7 anti-CD3, APC annexinV (Biolegend). UV LIVE/DEAD (ThermoFisher Scientific, # L23105). Labelled cells were sorted on a SORP-FACS-AriaII using a 70 m nozzle to 5 populations (see *Table 1*). Cell counts are reported in *Supplementary file 2*. Cells were analyzed using *FlowJo* (Tree Star) software.

## Library preparation for TCR-seq

All libraries in this work were prepared according to the published method (*Oakes et al., 2017*), with minor adaptations as described below. Briefly, total RNA was extracted from each of the seven populations using RNeasy Micro Kit (# 74004, Qiagen) and cleaned from excess DNA with DNAse 1 enzyme (# M6101, Promega). RNA samples were reverse transcribed to cDNA (SuperScript III, # 12574026, Invitrogen) using primers for the mouse α chain (mAlpha_RC2) and for the mouse β chain (mBeta_RC2) (see *Supplementary file 1*). Following reverse transcription the samples were purified on minielute spin columns (# 28004, QIAGEN). The cDNA was ligated to an oligonucleotide containing a unique 12 basepair molecular identifier (UMI) (6 N_I8.1_6 N_I8.1_SP2, see *Supplementary file 1*) using T4 RNA ligase (M0204S, NEB). Ligation products were purified using Agencourt AMPure XP beads (# A63881, BeckmanCoulter). Next, three rounds of extension PCR were executed (using KAPA HiFi DNA Polymerase, KAPA Biosystems) to add illumina sequencing adaptors and Illumina sample indices for multiplex sequencing (see *Supplementary file 2*). The thermal cycler parameters are an initial denaturation step (3 min at 95 °C) followed by cycles of denaturation (98 °C for 20 s), annealing (61 °C for 15 s), and extension 72 °C for 30 s. The final extension step was at 72 °C for 5 min. The lid was maintained at 105 °C. After the first round PCR (5 cycles), PCR products were purified using Agencourt AMPure XP beads and split in two, and α and β TCR genes were processed separately in subsequent steps. After the second PCR (8 cycles), PCR products were again purified using Agencourt AMPure XP beads. The final amplification using the adapter sequences P5 and P7 were carried out on a real-time qPCR machine, and the amplification was tracked by the incorporation of SYBR green. The

cycler was stopped manually when the fluorescent signal reached a predetermined threshold, thus preventing overamplification.

The final library concentration was measured using Qubit Fluorometric Quantification (ThermoFisher Scientific) and the presence of the correct 600–700 bp product confirmed by electrophoresis on a High Sensitivity D1000 ScreenTape cassette using a 4200 TapeStation System (Agilent). Multiple samples were pooled in equal molarity, and then sequenced using NextSeq 550 (200 bp forward read, 100 bp reverse) (Illumina).

## Pre-processing and error correction for raw reads

Data were processed using an in-house pipeline, coded in *R*. First, UMI sequences were transferred from read 2 to read 1. Trimmomatic was used to filter out the raw reads containing bases with Q-value ≤ 20 and trim reads containing adaptors sequences (*Bolger et al., 2014*). The remaining reads were separated according to their barcodes and reads containing the constant region for α or β chain primers sequences were filtered (CAGCAGGTTCTGGGTTCTGGATG / TGGGTGGAGTCACATTTCTC AGATCCT α and β chain, respectively), allowing up to three mismatches. To correct for possible sequence errors, we cluster the sequences UMIs' in two steps; (1) The UMIs with the highest frequency are grouped within a Levenshtein distance of 1 (*Levenshtein, 1966*). (2) Out of these sequences, CDR3AA sequences (starting from the most frequent sequence in a group) were clustered using a Hamming distance threshold of 4 (*Hamming, 1950*). Finally, the UMI of each CDR3 sequence was counted.

## Annotation and generation model

From the raw nucleotide reads, we performed a preliminary annotation using the python module *PyIR* (version 1.3.0) (*Soto et al., 2020*), which provides a wrapper and parser of the open source software *IgBlast* (*Ye et al., 2013*). We then separated the productive clonotypes from the out of frame reads and/or reads containing stop codons. We define a clonotype as TCRs sharing V genes, J genes, and the same CDR3 nucleotide sequence. If different reads are annotated as the same clonotype in the same dataset, only the read with highest UMI counts is considered.

For our models, we use a reduced set of genes from the IMGT free online repository (*Lefranc et al., 2015*) in order to have a single allele per gene, preferring functional alleles to open reading frame or pseudo genes. A further reduction is done for the V genes of the α chain, clustering to a single representative all of the those genes that result indistinguishable in the region from the maximum observed V offset for the annotation to the conserved cysteine. Two genes are said to be indistinguishable if the Hamming distance (*Hamming, 1950*) between the considered regions is equal to 0. For each TRAV cluster, we choose as the representative the most frequent gene in the preliminary annotation. In this way, we obtain 76 V genes and 51 J genes for the α chain, 26 V genes, 2 D genes and 14 J genes for the β chain.

In order to infer a generation model, we use the open source software *IGoR* (*Marcou et al., 2018*) on all out-of-frame clonotypes pooled from all maturation stages of all mice. The generation model associates to each α (β) read a probability $P_{gen}$ of being generated through the VJ (VDJ) recombination process. After learning a generation model, we annotate the reads using the most probable alignment scenario using the IGoR software, as the clonotype (V, J gene choice, CDR3 nucleotide sequence) with the highest $P_{gen}$ among all possible recombination scenarios.

The PCA was computed in *R* (version 3.6.0) using the function "PCA" from the *FactoMineR* package (version 2.4).

## Statistical classification

The features are assigned to each α chain as a binary vector $\vec{\sigma}$, where each entry is equal to 1 if the feature is observed, 0 otherwise. In this study the set of features is encoded using the 'SoniaLeftposRightpos' class (from the Python package *Sonia* version 0.0.45) which provides 5033 features: 30 for the CDR3 amino acid lengths, 25 left to right positions for each of the 20 amino acids (500 features), 25 right to left positions (500), the joint V/J gene usage (76×51 = 3876) and the independent usage (76+51 = 127). Analogously for the β we obtain 1434 features (without considering D genes).

To learn the models for the statistical classification of two stages, we first remove all sequences that share the same features between the two sets (i.e. same amino acid CDR3, V and J gene). Then,

we balance the size of the sets sub-sampling the larger one so that its size does not exceed 25% of the size of the smaller. Each of the resulting sets is divided into a train and a test set by a ratio 70%/30% ('StratifiedShuffleSplit', module 'model_selection' from the Python package *scikit-learn*, version 0.24.2). The classifiers are learned with linear models, defined by a single layer with binary cross entropy as a loss function, binary accuracy as metrics, a sigmoid as activation function, coded using the 'keras' module from the Python package *tensorflow* (version 2.4.1). We obtained similar performance for the classification task by learning with a random forest algorithm as provided by the function 'RandomForestModel' in the module 'keras' from the package *tensorflow_decision_forests* (version 0.2.4).

## Selection model

To learn a $P_{\text{stage}}$ selection model for each maturation stage, we pooled together the annotated sequences from all mice for the given maturation stage, discarding all clonotypes annotated with non-functional and pseudo genes. We learn a selection model using the open source software *Sonia* for each maturation stage. *Sonia* performs a linear regression over the features of the sequences in the dataset to infer the enrichment ratio between the maturation specific dataset and the generation model. The feature choice for the enrichment model is similar, except for the fact that only independent gene usage is considered, reducing features to 1157 for α chain (1070 for β chain). The probability of observing a sequence in a stage is modeled as

$$P_{\text{stage}}(\vec{\sigma}) = \frac{1}{Z} e^{-E_{\text{stage}}(\vec{\sigma})} P(\vec{\sigma}) \tag{1}$$

where $Z$ is a normalization factor and the energy $E_{\text{stage}}(\vec{\sigma})$ for a sequence showing a set of features $\mathcal{F}(\vec{\sigma})$ is defined as

$$E_{\text{stage}}(\vec{\sigma}) = \sum_{f \in \mathcal{F}(\vec{\sigma})} \epsilon_{\text{stage}}^{(f)}(\vec{\sigma}) \tag{2}$$

Here $\epsilon^{(f)}$ is a weight associated to the feature $f$ and is learnt from data. To look at specific enhanced features between stages $a$ and $b$ one can obtain the average weights difference from the respective $P_{\text{stage}}$ models as

$$\left\langle \epsilon_a^{(f)} - \epsilon_b^{(f)} \right\rangle = \frac{p_a^{(f)} + p_b^{(f)}}{2} \cdot \left( \epsilon_a^{(f)} - \epsilon_b^{(f)} \right) \tag{3}$$

where $p_{\text{stage}}^{(f)}$ is the marginal associated by the model to the feature.

The limited amount of clonotypes for certain maturation stages precludes using deep neural network based selection models, although we do not expect the conclusions to change with the *DNN SoNNia* model *Isacchini et al., 2021*.

## Hydrophobicity score

To study the hydrophobicity increase with respect to the generation, we define a stage-wide score as

$$\text{U} = \sum_{\substack{a \in \text{hydro} \\ x \in \text{CDR3cr}}} \epsilon^{(a|x)} \cdot p_{\text{gen}}^{(a|x)} \tag{4}$$

where $\epsilon^{(a|x)}$ is the weight associated by the model to the amino acid $a$ at position $x$; the marginal $p_{\text{gen}}^{(a|x)}$ is obtained by the generation model on the same feature (see previous sections). The sum runs over the hydrophobic amino acids CFILMWY, following the definition from *Lagattuta et al., 2022*, considering just the positions of our model which correspond to the central region p108-p114 of the CDR3 in IMGT convention (model positions (4:10) from the left, and, (–11:–5) from the right). We also define an index $u$ for hydrophobicity which can now be associated to each sequence as follow

$$\text{u} = \sum_{\substack{a \in \text{hydro} \\ x \in \text{CDR3cr}}} \frac{1}{L} \tag{5}$$

that is the number of hydrophobic residues found in the central region (same choices as above), normalized by the CDR3 length $L$.

## *n*-gram Shannon entropy estimation

As a diversity measure we consider the Shannon entropy defined as:

$$S = - \sum_i p(i) \log_2 p(i) \tag{6}$$

where $p(i)$ is the probability of finding a clonotype in the data. Since $n$-grams are sampled from $20^n$ possible motifs, undersampling could bias a naive estimation of the entropy. We overcome this bias by estimating the Shannon entropy using the Nemenman-Shafee-Bialek (NSB) estimator (**Nemenman et al., 2002**). The NSB estimator is computationally tractable and calculates an estimation error. We implement the entropy and variance estimators as given in **Archer et al., 2014**. We verified the NSB estimators better performance for our datasets compared to other non-parametric estimators (**Figure 4—figure supplement 1A, B**), consistently with previous reports (**Archer et al., 2014**). To check for convergence we subsample the clonotypes in the dataset at increasing sizes and estimate the entropy for each sub-sample (**Figure 4—figure supplement 2**). Convergence sets a limit legnth of $n = 4$ due to sample size constraints of the smallest dataset.

## Full model Shannon entropy estimation

The Shannon entropy in **Equation 6** associated to the full $p = P_{\text{stage}}(\vec{\sigma})$ model requires summing over all possible clonotypes $i = \vec{\sigma}$. Practically we evaluate the entropy by producing synthetic sequences according to the selection model $P_{\text{stage}}$ and averaging the value of $\log_2 P_{\text{stage}}$.

$$S(P_{\text{stage}}) \simeq \frac{1}{N} \sum_{k=1}^{N} \log_2 P_{\text{stage}}(\vec{\sigma}_k^*) \tag{7}$$

with clonotypes $\vec{\sigma}_k^*$ sampled from the $P_{\text{stage}}$ distribution.

Because of sequencing errors, the entropy of $n$-grams is systematically overestimated in the data. To estimate and correct for this bias, we measured the error rate from the data, provided as a byproduct of the IGoR training procedure (**Marcou et al., 2018**). We used this rate to produce synthetic sequences with simulated sequencing errors. The difference in $n$-gram entropy between error-prone and error-free sequences was then applied as a subtractive correction factor to the data.

## *n*-gram Jensen-Shannon divergence

To quantify the distance between two distributions $p_a$ and $p_b$ defined on the same support, we use the symmetric Jensen-Shannon divergence JSD:

$$\text{JSD}\left(p_a, p_b\right) = \frac{1}{2} \sum_i p_a(i) \log_2 \frac{2p_a(i)}{p_a(i)+p_b(i)} + \left(a \leftrightarrow b\right) \tag{8}$$

where the sum runs over all possible observables $i$ and the term $\left(a \leftrightarrow b\right)$ corresponds to the same expression in the first one with $a$ and $b$ inverted. The Jensen-Shannon divergence is bounded between 0 and 1 bits, with JSD = 0 bits if the distributions are identical and a maximal difference of JSD = 1 bit. We use JSD to asses the divergence between $n$-gram distributions and between selection models.

## Full model Jensen-Shannon divergence

To compare selection models of complete clonotypes at two maturation stages, the divergence between the $P_{\text{stage}}$ distribution of model $a$ and the model $b$ is:

$$\text{JSD}\left(P_a, P_b\right) \simeq \frac{1}{2N} \sum_{k=1}^{N} \log_2 \frac{2e^{-E_a(\vec{\sigma}_k^a)}}{e^{-E_a(\vec{\sigma}_k^a)}+e^{-E_b(\vec{\sigma}_k^a)}} + \left(a \leftrightarrow b\right) \tag{9}$$

where the clonotypes $\vec{\sigma}_k^a$ are sampled from the $P_a$ distribution. In **Equation 9**, we used the fact that a given sequence has the same background generation probability $P_{\text{gen}}$ in both selection models.

## Discrimination in the thymus vs discrimination in the periphery

Here we show a formal link between discramation of negatively selected vs non-negatively selected TCR on the one hand, and foreign vs self-peptide recognition on the other.

We start by considering (negative) thymic selection. Following *Butler et al., 2013*, we assume that a random TCR will recognize any peptide with probability $p$. Then the number of recognized self-peptides $n$ by a random TCR is distributed according to a Poisson law of mean $\bar{n} = pN$, where $N$ is the number of self-peptides, $P(n) = \text{Poiss}[pN](n) \equiv e^{-pN}(pN)^n/n!$.

If each TCR only screens $M = fN$ self-peptides, with $f < 1$, then the probability of passing selection (and ending up in spleen) is $P(\text{spleen}|n) = (1 - n/N)^M \approx e^{-nf}$, and $P(\text{apo}|n) = 1 - e^{-nf}$ for the probability of ending up in apo (as apostosis, i.e. single-positive cells expressing Annexin V as in our experiments).

We assume that the discriminator of apo vs spleen single positives is perfect, in the sense it can perfectly deduce $n$ from the TCR sequence. In this idealized case, discrimination errors are entirely attributable to the partial screening of self-peptides. Using Bayes' law, one can show that the distributions of $n$ in spleen and apo read:

$$P(n|\text{spleen}) \quad = \frac{P(\text{spleen}|n)P(n)}{P(\text{spleen})} = \text{Poiss}[pN(1-f)](n), \tag{10}$$

$$P(n|\text{apo}) \quad = \frac{P(\text{apo}|n)P(n)}{P(\text{apo})} = \frac{(1-e^{-nM/N})(pN)^n e^{-pN}}{n!(1-e^{-pM})} \approx \frac{nf}{pNf}\frac{(pN)^n e^{-pN}}{n!} = \text{Poiss}[pN](n-1), \tag{11}$$

where the first equation results from direct algebra, and the second is obtained in the limit of small $f$. The AUC of the discrimination task is then given by the probability that drawing a random number from a Poisson of mean $\bar{n}(1-f)$ yields a smaller number than drawing a random number from a Poisson of mean $\bar{n}$, and adding 1 to it. If we now use the observation of $m$ TCRs from the same subset (apo or spleen) instead of a single one, the task becomes easier: We can form a collective score by adding up the $n$'s of each TCR (since they are independent draws from either the apo or spleen ensembles) so that the two Poisson distributions, of respective means $m\bar{n}(1-f)$ and $m\bar{n}$, become better separated. This is qualitatively the result of *Figure 3B*, which is based on the learned score, rather than on an idealized one.

We now turn to the case of self vs foreign peptide discrimination by a group of $R$ T cells recruited to a site of infection. If the peptide is from the self, then the probability for a given circulating TCR to recognize it is $p(1-f)$ (*Butler et al., 2013*). Then the number of specific TCR is Poisson distributed with mean $p(1-f)R$. If the peptide is foreign, that number is also Poisson distributed, but with mean $pR$. Again, the AUC of the discrimination task is given by the probability of drawing a smaller number from the former distribution than from the latter. This task is expected to be at least as hard as that of apo vs spleen TCR discrimination when $pR \approx m\bar{n}$, where $pR$ is the expected number of TCR specific to the foreign antigen.

## Other software for statistical analysis

The Jensen-Shannon dendrograms linkage is computed by the Ward method as provided by the function 'linkage', reordered according to the function 'optimal_leaf_ordering', both from the Python module 'cluster.hierarchy' in *scipy* package (version 1.7.3). The Pearson correlation coefficient is computed with the Python function 'pearsonr' as contained in the module 'stats' in the *scipy* package. The coefficient of determination $R^2$ is computed with the Python function 'r2_score' as contained in the module 'metrics' in the *sklearn* package.

## Code availability

All code for reproducing the figures of this paper can be found at https://github.com/statbiophys/thymic_development_2022, (copy archived at swh:1:rev:ce4966eee8937544d8c97ff956a049f-c32d279da; *Camaglia, 2022*).

## Acknowledgements

The study was supported by a '80 prime' CNRS-Weizmann PhD scholarship, European Research Council COG 724208 and ANR-19-CE45-0018 'RESP- REP' from the Agence Nationale de la Recherche and DFG grant CRC 1310 'Predictability in Evolution'.

## Additional information

### Competing interests
Aleksandra M Walczak: Senior editor, eLife. The other authors declare that no competing interests exist.

### Funding

| Funder | Grant reference number | Author |
|---|---|---|
| CNRS-Weizmann | 80 prime | Francesco Camaglia |
| European Research Council | COG 724208 | Aleksandra M Walczak |
| Agence Nationale de la Recherche | ANR-19-CE45-0018 | Thierry Mora |

The funders had no role in study design, data collection and interpretation, or the decision to submit the work for publication.

### Author contributions
Francesco Camaglia, Conceptualization, Data curation, Software, Formal analysis, Validation, Investigation, Visualization, Methodology, Writing – original draft, Writing – review and editing; Arie Ryvkin, Data curation, Validation, Investigation, Methodology, Writing – review and editing; Erez Greenstein, Conceptualization, Resources, Data curation, Formal analysis, Validation, Investigation, Visualization, Methodology, Writing – review and editing; Shlomit Reich-Zeliger, Resources, Supervision, Methodology; Benny Chain, Supervision, Writing – original draft, Project administration, Writing – review and editing; Thierry Mora, Conceptualization, Resources, Formal analysis, Supervision, Funding acquisition, Investigation, Visualization, Methodology, Writing – original draft, Project administration, Writing – review and editing; Aleksandra M Walczak, Conceptualization, Resources, Formal analysis, Supervision, Funding acquisition, Investigation, Methodology, Writing – original draft, Project administration, Writing – review and editing; Nir Friedman, Conceptualization, Resources, Data curation, Supervision, Funding acquisition, Validation, Investigation, Methodology, Project administration

### Author ORCIDs
Francesco Camaglia ⓘ http://orcid.org/0000-0003-4810-2631
Erez Greenstein ⓘ http://orcid.org/0000-0002-6923-8469
Benny Chain ⓘ http://orcid.org/0000-0002-7417-3970
Thierry Mora ⓘ http://orcid.org/0000-0002-5456-9361
Aleksandra M Walczak ⓘ http://orcid.org/0000-0002-2686-5702

### Ethics
The experiment was carried out using three 6-weeks old male inbred Nur77-GFP/Foxp3-mCherry (C57BL/6 background). The cross was bred and maintained at the Weizmann institute. All animals were handled according to Weizmann Institute's Animal Care guidelines, in compliance with national and international regulations. This study was performed in strict accordance with the recommendations in the Guide for the Care and Use of Laboratory Animals of the National Institutes of Health. All of the animals were handled according to approved institutional animal care and use committee (IACUC) protocols (#21661115-2) of the Weizmann Institute of Science. The protocol was approved by the Committee on the Ethics of Animal Experiments of the Weizmann Institute of Science. Every effort was made to minimize suffering.

### Decision letter and Author response
Decision letter https://doi.org/10.7554/eLife.81622.sa1
Author response https://doi.org/10.7554/eLife.81622.sa2

## Additional files

### Supplementary files
- MDAR checklist
- Supplementary file 1. Primers list.
- Supplementary file 2. Cell counts.

### Data availability
All code for reproducing the figures of this paper can be found at https://github.com/statbiophys/thymic_development_2022.git, (copy archived at swh:1:rev:ce4966eee8937544d8c97ff956a049f-c32d279da). The data has been deposited on the SRA as BioProject ID PRJNA804508 http://www.ncbi.nlm.nih.gov/bioproject/804508.

The following dataset was generated:

| Author(s) | Year | Dataset title | Dataset URL | Database and Identifier |
|---|---|---|---|---|
| Camaglia F, Ryvkin A, Greenstein E, Reich-Zeliger S, Benny Chain TM, Aleksandra MW, Nir F | 2022 | thymic selection repertoires mice | http://www.ncbi.nlm.nih.gov/bioproject/PRJNA804508 | NCBI BioProject, PRJNA804508 |

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
