## [Editor Report]

This paper addresses an important question within adaptive immunity, namely whether the T cell receptor (TCR) repertoire of negatively selected thymocytes shares common features. The authors analyze T cell receptor sequences from mice as they progress through positive selection, CD4/CD8 lineage commitment, and negative selection. Thereby they find small but consistent differences between the repertoires at these selection stages, providing arguments that their findings do not indicate any sequence-specific selection.

---

## [Decision Letter]

**Decision letter after peer review:**

Thank you for submitting your article "Population based selection shapes the T cell receptor repertoire during thymic development" for consideration by *eLife*. Your article has been reviewed by 3 peer reviewers, one of whom is a member of our Board of Reviewing Editors, and the evaluation has been overseen by Tadatsugu Taniguchi as the Senior Editor. The reviewers have opted to remain anonymous.

All reviewers appreciated the combination of an interesting data set with sophisticated analytical methods to address the very relevant question of how T cell selection is regulated. However, there is the consensus impression that some of the claims and interpretations are not fully supported by the presented analyses. This also affects the main message of the manuscript put prominently within the title, which is one possible hypothesis, but not the only explanation for the presented observations.

In the light of the comments, we would be willing to consider a substantially revised version of the manuscript that addresses all the points mentioned by the reviewers, and which will be subject to a re-review. In particular, the revision should address the following main recommendations:

Essential revisions:

1) A careful revision of each of the claims and conclusions. You should discuss/re-consider for each of the analyses the limitations that are associated with the methods and the underlying data, affecting statistical evidence. In particular this relates to the analysis presented in Figure 3 and 5 (see comment reviewer 3), with especially the former being insufficient in supporting the performed interpretations.

2) Replacement of the title and careful restructuring of the text to better match supported conclusions and to motivate the performed analyses (see specific comment reviewer 2).

3) Discussion of the used experimental system (Nur77EGFP/Annexin V) and the possibility to detect possible differences within a data set of the presented size.

Please also see the detailed comments of the individual reviewers below. As these comments would require substantial changes to the manuscript, partly affecting its central message, please note that a revised manuscript will be subject to a re-review.

*Reviewer #1 (Recommendations for the authors):*

1) The authors mention that they generally pool the read outs from three mice to perform the repertoire comparison, as e.g. based on the n-gram distributions, to increase the statistical power. Despite the reasonable expectation of mice similarity, the questions remains if there might be stronger signals for selection with development stage on an individual level, and the individual signal possibly being masked by the pooled analysis. Looking at Figure S3 with the statistics for the individual mice, there seem to be some (minor) differences in the number of readouts per differentiation step between mice that could affect the analysis. CDR3 length distribution (Figure 2B) and n-gram analysis could be performed on an individual mouse level to provide evidence for the comparability of readouts of individual mice. I assume that e.g. Figure 2B is based on the pooled mouse data. I think it would be helpful to perform some statistical analyses on the level of individual mice to corroborate their similarity and appropriateness for pooling.

2) Another point concerns the fact that α- and β-chain are analyzed separately, and not in combination. This could also mask possible signatures as already indicated and discussed by the authors. To which extent do the acquired data allow a combined analysis of α- and β-chains with regard to a "connected selectivity"?

3) The suitability of n-gram distributions and the specific selection of n for comparison might also be dependent on the actual CDR3 length, i.e. only sequences with a sufficient CDR3-length could contain information for appropriate n-gram comparisons. I assume that the n-gram comparison considered all sequences with a CDR3-length of at least n. I am not an expert and this might be an unreasonable suggestion, but would it make sense to compare the n-gram distribution for one particular CDR3-length (e.g. 12) to determine if there are differences between differentiation steps? I think such an approach would assume that there are differences in selection due to the CDR3-length and n-gram distribution, while the current approach neglects the CDR3-length.

*Reviewer #2 (Recommendations for the authors):*

The following points should be addressable with edits of the text:

1. The sorting approach should be discussed with pros and cons. A critical question is the lifetime of negatively selected cells after they received their instruction to go into apoptosis. If strong TCR signal strength translates into short subsequent lifetime, then the annexin+ cells sorted may be enriched for cells receiving a rather weaker negative selection signal. This caveat should be discussed. The sorting approach as such is valid as thymocytes destined to die are identified.

2. Kyewski and co-workers have found that 1-3% of thymic epithelial cells robustly present any given peptide they tested (using unrelated peptides from proteins encoded on different chromosomes), sometimes even a higher fraction, 5%; see Pinto et al. PNAS 2013 and references therein. Hence, a thymocyte encountering of the order of 100 mTECs could sample practically the entire self-peptidome. Le Borgne, M. et al. (Ref. 7 in the manuscript) give the actual number scanned as up to 500. Hence, the number of distinct peptides seen by a thymocyte could easily be much higher than the 1-5% of the self-peptidome given by the authors (Le Borgne et al., cited in evidence of this low percentage, do not give such a number, to my knowledge, while Ref. 6 proposes a very interesting theoretical scenario, i.e., if the fraction were so small, selection could still work by quorum sensing, but gives no experimental evidence). Of course, the actual fraction of peptides seen is not clear, as it will depend on the spatial distribution of peptides on mTECs, the path taken by thymocytes, and the sensitivity of recognition. However, the upper limits allowed by the available experimental data appears to be the majority, if not close to 100%, of self-peptides, and these data should be discussed with due balance. Hence my request to adapt this aspect in Introduction, Discussion and Title accordingly.

3. All TCR α results should be shown in parallel for β (possibly using supplementary Figures, as the authors have done already to a large extent).

4. A point in the Intro not related to the message of the manuscript: "… imposes.… boundaries on the binding energy.…" The TCR senses also koff, a kinetic property. Beyond a host of indirect evidence, this was demonstrated directly by two optogenetics papers in *eLife* in 2019 (Yousefi et al.; Tischer and Weiner).

*Reviewer #3 (Recommendations for the authors):*

– I would suggest making explicit which kind of changes to the repertoire this study is designed (or able) to detect, and which kind of changes it cannot detect.

– Related to the above, the limitations of the study could be more clearly emphasized, and some of the conclusions might be somewhat toned down.

– The data shown in Figure 3B is misleading, as explained above; the authors should consider removing this analysis entirely but at least the interpretation of these data as some sort of evidence for collective decision-making by T cells should be removed.

– In Figure 4, is it truly necessary to consider 3-grams? Is the discriminatory power of 3-grams (or shorter k-grams) better than that of the P_stage_ models and if so, might this indicate overfitting/too high model complexity of the P_stage_ models?

– The scenario simulated in Figure 5A might be too different from the one analyzed in Figure 5B to be a meaningful comparison. The general idea of the negative correlation is obviously valid but can be illustrated using a much simpler model (or even just the basic mathematical argument); alternatively, the model could also incorporate common causes of selection to be more useful.

– Page 1: The ability to discriminate between self and non-self targets "cannot therefore be inherited" – it could be possible that V and J segments themselves evolved to make receptors that are biased towards recognizing non-self targets?

– Page 2, left: "cross-reactive to an overwhelming majority of self- peptides." – unclear what this means, surely a single TCR can't recognize a majority of self-peptides?

– Page 2, left: discussion of the populations. Here it could already been mentioned that the DPpos population might contain cells that still get negatively selected later, as mentioned in the Discussion.

– Page 2, bottom right: the pooling subtly but importantly changes the meaning of the performed inferential statistics. (What is the experiment repetition in the frequentist framework?)

– Figure 1: Why was it necessary to learn a new generation model for these sequences rather than use an existing (mouse-based) one?

– Figure 1: The Annexin V+ populations are small, and the gating looks somewhat arbitrary, which could be mentioned as a limitation.

– Page 4, bottom left: What about the distribution of N nucleotides rather than the length of the entire sequence?

– Page 5, left: "even the smallest AUC are highly significant" – this is just a consequence of the large number of sequences and a low p-value does not make a small AUC more important

– Page 5, left: "Thus statistical properties of a repertoire can distinguish it from another repertoire, even when the feature distributions of individual TCRs are largely overlapping." – you could mention that this is just the law of large numbers

– Page 5, top right: NSB entropy estimator – did this matter for your results?

– idem "The maximum possible entropy if all 20 amino acids were used uniformly would be S/n = log2 20 ∼ 4.3 bits." – it is known that amino acid is non-uniform, this is the case for all proteins. A more meaningful baseline would be the entropy for the average frequencies of amino acids

– Page 6, left: "there is no evidence that selection at any step involves complex sequence motifs, or amino acid interactions, which would not be captured by the linear model" – It is not clear why the presence of such "motifs" would be expected in the first place, but there could still be strong selection against TCRs that recognize peptides in certain (ubiquitous) proteins, and this would not necessarily translate to simple "motifs" that are readily detectable

– idem "validating the hypothesis of linear selection factors" – I am not sure if this is meant to imply that such linear selection factors truly drive selection. It would seem more likely that the observed differences are population-level consequences of a selection process that operates in a (stochastic) sequence-specific manner

– Discussion "Consistent with these previous results, we can project our model parameters to build a single hydrophobicity index, which we observe to be significantly increased in positively selected cells (DP pos) versus DP pre, and decreased in single positive sets (Figure S21)." – this raises the question whether the single hydrophobicity index might already be sufficient to explain the differences seen in the repertoire

– "This number is in agreement with our observations (Figure 5B) that ∼ 30 TCRs increases the AUC of correctly classifying these cells as CD4^+^ spleen vs CD4 apo from ∼ 0.59 to ∼ 0.87." – I think this is not a valid interpretation. First, as noted above, this phenomenon would be observed for any such classification. Second, there is nothing special to an AUC value of 0.87

– Methods: The Adam optimizer was used for the linear classifiers. Was this necessary (e.g., were the models underspecified) or could a deterministic optimizer have been used?

[Editors' note: further revisions were suggested prior to acceptance, as described below.]

Thank you for resubmitting your work entitled "Population based selection shapes the T cell receptor repertoire during thymic development" for further consideration by *eLife*. Your revised article has been evaluated by Tadatsugu Taniguchi (Senior Editor) and a Reviewing Editor. In addition, the previous reviewers have also assessed the submission

You performed a substantial amount of work to address the previous comments and the manuscript has been substantially improved. However, there are some remaining issues that need to be addressed, as outlined below:

In particular, the previously identified misinterpretation of "classification of T cells" as "quorum sensing by T cells" is partly re-inforced by the added changes, especially within the Results section (see comment reviewer 3). We would therefore ask you to carefully revisit the corresponding paragraphs within the results and Discussion section and rephrase where necessary. See also the specific recommendations of the reviewer below.

*Reviewer #2 (Recommendations for the authors):*

The authors have done an excellent job of revising the manuscript. I recommend publication as it is.

*Reviewer #3 (Recommendations for the authors):*

The authors have made many changes to the manuscript and added additional findings as well. I think the manuscript has been improved in many places. However, I suspect the revision is not complete. My main criticism had been the interpretation of imperfect classification as evidence for a quorum sensing mechanism. The authors state in their rebuttal letter that they agree with me and that they "radically rewrote" the corresponding text. However, the relevant paragraph in the discussion, the one starting with "Overall, the statistical population-level differences between populations of thymocytes and mature T cells.…", is entirely unchanged from the previous version. There have been some additional explanations inserted in the Results section, but these are not really correct (i.e. it is said that T cells are "weak learners", but the "learner" in this context is the classification algorithm built by the authors, not the T cell, which is the instance being classified.) So the same interpretation mistakes are unfortunately still present in the revised version. Perhaps the authors overlooked something.

---

## [Author Response]

Essential revisions:1) A careful revision of each of the claims and conclusions. You should discuss/re-consider for each of the analyses the limitations that are associated with the methods and the underlying data, affecting statistical evidence. In particular this relates to the analysis presented in Figure 3 and 5 (see comment reviewer 3), with especially the former being insufficient in supporting the performed interpretations.

We have carried out a careful revision taking all these points in consideration. In particular we have thoroughly revised the interpretation of *Figure 3* in accordance with the comments of reviewer 3, which we completely agree with. We have kept in the “toy” example in *Figure 5*, because we feel that this greatly clarifies the method for the non-specialist reader. But we have addressed the point regarding the interpretation, by reorganising the figure with the additional analyses suggested as specified below.

2) Replacement of the title and careful restructuring of the text to better match supported conclusions and to motivate the performed analyses (see specific comment reviewer 2).

We have changed title and addressed the motivation as raised by reviewer 2.

3) Discussion of the used experimental system (Nur77EGFP/Annexin V) and the possibility to detect possible differences within a data set of the presented size.

We have introduced new text at various points in the manuscript, highlighting the limitations of the methods.

Reviewer #1 (Recommendations for the authors):1) The authors mention that they generally pool the read outs from three mice to perform the repertoire comparison, as e.g. based on the n-gram distributions, to increase the statistical power. Despite the reasonable expectation of mice similarity, the questions remains if there might be stronger signals for selection with development stage on an individual level, and the individual signal possibly being masked by the pooled analysis. Looking at Figure S3 with the statistics for the individual mice, there seem to be some (minor) differences in the number of readouts per differentiation step between mice that could affect the analysis. CDR3 length distribution (Figure 2B) and n-gram analysis could be performed on an individual mouse level to provide evidence for the comparability of readouts of individual mice. I assume that e.g. Figure 2B is based on the pooled mouse data. I think it would be helpful to perform some statistical analyses on the level of individual mice to corroborate their similarity and appropriateness for pooling.

We agree that analysis of individual mice could give additional information. We have redone *Figure 2* analyses to include individual mice data. However, at a TCR population level the intermouse effects are very small, and the results are not influenced by variability in TCR count between mice.

2) Another point concerns the fact that α- and β-chain are analyzed separately, and not in combination. This could also mask possible signatures as already indicated and discussed by the authors. To which extent do the acquired data allow a combined analysis of α- and β-chains with regard to a "connected selectivity"?

Our ability to co-analyse α and β chains is limited because the method we use does not pair individual TCR α and β chains. So the α and β sequence data are effectively independent data sets.

3) The suitability of n-gram distributions and the specific selection of n for comparison might also be dependent on the actual CDR3 length, i.e. only sequences with a sufficient CDR3-length could contain information for appropriate n-gram comparisons. I assume that the n-gram comparison considered all sequences with a CDR3-length of at least n. I am not an expert and this might be an unreasonable suggestion, but would it make sense to compare the n-gram distribution for one particular CDR3-length (e.g. 12) to determine if there are differences between differentiation steps? I think such an approach would assume that there are differences in selection due to the CDR3-length and n-gram distribution, while the current approach neglects the CDR3-length.

We have done this analysis and included it in *Figure 4—figure supplement 9*. There were small differences in entropy when using different CDR3 length, but the differences are small. We therefore combined all length CDR3s for subsequent analyses to increase statistical power.

Reviewer #2 (Recommendations for the authors):The following points should be addressable with edits of the text:1. The sorting approach should be discussed with pros and cons. A critical question is the lifetime of negatively selected cells after they received their instruction to go into apoptosis. If strong TCR signal strength translates into short subsequent lifetime, then the annexin+ cells sorted may be enriched for cells receiving a rather weaker negative selection signal. This caveat should be discussed. The sorting approach as such is valid as thymocytes destined to die are identified.

We agree with the reviewer. We have added a short paragraph discussing this point in the *Discussion*.

2. Kyewski and co-workers have found that 1-3% of thymic epithelial cells robustly present any given peptide they tested (using unrelated peptides from proteins encoded on different chromosomes), sometimes even a higher fraction, 5%; see Pinto et al. PNAS 2013 and references therein. Hence, a thymocyte encountering of the order of 100 mTECs could sample practically the entire self-peptidome. Le Borgne, M. et al. (Ref. 7 in the manuscript) give the actual number scanned as up to 500. Hence, the number of distinct peptides seen by a thymocyte could easily be much higher than the 1-5% of the self-peptidome given by the authors (Le Borgne et al., cited in evidence of this low percentage, do not give such a number, to my knowledge, while Ref. 6 proposes a very interesting theoretical scenario, i.e., if the fraction were so small, selection could still work by quorum sensing, but gives no experimental evidence). Of course, the actual fraction of peptides seen is not clear, as it will depend on the spatial distribution of peptides on mTECs, the path taken by thymocytes, and the sensitivity of recognition. However, the upper limits allowed by the available experimental data appears to be the majority, if not close to 100%, of self-peptides, and these data should be discussed with due balance. Hence my request to adapt this aspect in Introduction, Discussion and Title accordingly.

We agree with the reviewer. We have added a short paragraph discussing this point in the *Discussion*.

3. All TCR α results should be shown in parallel for β (possibly using supplementary Figures, as the authors have done already to a large extent).

We have added all the equivalent β results.

4. A point in the Intro not related to the message of the manuscript: "… imposes.… boundaries on the binding energy.…" The TCR senses also koff, a kinetic property. Beyond a host of indirect evidence, this was demonstrated directly by two optogenetics papers in eLife in 2019 (Yousefi et al.; Tischer and Weiner).

We agree and have edited the text accordingly.

Reviewer #3 (Recommendations for the authors):– I would suggest making explicit which kind of changes to the repertoire this study is designed (or able) to detect, and which kind of changes it cannot detect.

We have added a section to the *Discussion* which addresses these points.

– Related to the above, the limitations of the study could be more clearly emphasized, and some of the conclusions might be somewhat toned down.

We have added a section to the *Discussion* which addresses these points.

– The data shown in Figure 3B is misleading, as explained above; the authors should consider removing this analysis entirely but at least the interpretation of these data as some sort of evidence for collective decision-making by T cells should be removed.

We are grateful to the reviewer for picking up this important point. The concept of weak learners, i.e. each TCR cannot be unambiguously assigned to a class but that an ensemble of TCRs can, is exactly the point we are trying to make in *Figure 3*. We agree that there is a big jump from this to saying the mechanisms of tolerance in the periphery is due to quorum sensing. This is just one possible strategy to solve the problem of how the immune system makes the self/non-self decision if selection is leaky. We have substantially re-written the text to clarify this.

– In Figure 4, is it truly necessary to consider 3-grams? Is the discriminatory power of 3-grams (or shorter k-grams) better than that of the P_stage_ models and if so, might this indicate overfitting/too high model complexity of the P_stage_ models?

We have directly compared the discriminatory power of ngram=3 versus ngram=1 (amino acid usage) in *Figure 4—figure supplement 11*, and find that the ngram=3 model outperforms the single amino acid model in almost all comparisons.

– The scenario simulated in Figure 5A might be too different from the one analyzed in Figure 5B to be a meaningful comparison. The general idea of the negative correlation is obviously valid but can be illustrated using a much simpler model (or even just the basic mathematical argument); alternatively, the model could also incorporate common causes of selection to be more useful.

We disagree with the reviewer about our !toy” model which we feel makes the approach much more understandable to the general reader. However, we agree that common cause of selection is an important confounder, thus we decided to add another example related to hydrophobicity to provide further intuition on the expected correlations.

We also reorganised *Figure 5* in response to the reviewer"s comment in the public review, reporting only the most significant results in the manuscript and moving the rest in the supplementary material. For the correlations between the relative enrichments we use now DP stages relative to DP pre and SP stages relative to DP pos as suggested. The general message is unchanged.

– Page 1: The ability to discriminate between self and non-self targets "cannot therefore be inherited" – it could be possible that V and J segments themselves evolved to make receptors that are biased towards recognizing non-self targets?

We agree and have edited the text appropriately.

– Page 2, left: "cross-reactive to an overwhelming majority of self- peptides." – unclear what this means, surely a single TCR can't recognize a majority of self-peptides?

This sentence has been removed in the new text.

– Page 2, left: discussion of the populations. Here it could already been mentioned that the DPpos population might contain cells that still get negatively selected later, as mentioned in the Discussion.

We have added a sentence to this effect.

– Page 2, bottom right: the pooling subtly but importantly changes the meaning of the performed inferential statistics. (What is the experiment repetition in the frequentist framework?)

We have carried out additional analyses on individual mice, as specified in response to reviewer, and obtained similar results as on pooled data (*Figure 3—figure supplement 5*).

– Figure 1: Why was it necessary to learn a new generation model for these sequences rather than use an existing (mouse-based) one?

The generation model varies subtly, but significantly depending on mouse strain. Since we used a newly constructed transgenic, we felt it was safer to learn a new generation model. In addition, at the time of the article no model was available for the mouse α chain TCRs.

– Figure 1: The Annexin V+ populations are small, and the gating looks somewhat arbitrary, which could be mentioned as a limitation.

We have added this to the *Results section*.

– Page 4, bottom left: What about the distribution of N nucleotides rather than the length of the entire sequence?

For individual sequences it is not possible to assign the N nucleotides deterministically and thus they are in fact hidden variables [Marcou, 2018. Nature Communications]. We therefore prefer to use the observed variable of CDR3 length, which is more directly linked to the biology the selection process, rather than the N nucleotide distribution which remains hidden to selection.

– Page 5, left: "even the smallest AUC are highly significant" – this is just a consequence of the large number of sequences and a low p-value does not make a small AUC more important

We agree and have removed this sentence.

– Page 5, left: "Thus statistical properties of a repertoire can distinguish it from another repertoire, even when the feature distributions of individual TCRs are largely overlapping." – you could mention that this is just the law of large numbers

While we agree that overlapping populations can be statistically separated, and that the larger the number of observations, the more precisely and significantly the underlying populations can be identified (the law of large numbers) we think that specifically referring this law may confuse the non-mathematical reader. But we have radically re-written the manuscript to make this point much clearer.

– Page 5, top right: NSB entropy estimator – did this matter for your results?

We have enriched the analysis comparing different estimators where we show that the NSB estimator worked best (*Figure 4—figure supplement 8* and *Figure 4—figure supplement 9*).

– idem "The maximum possible entropy if all 20 amino acids were used uniformly would be S/n = log2 20 ∼ 4.3 bits." – it is known that amino acid is non-uniform, this is the case for all proteins. A more meaningful baseline would be the entropy for the average frequencies of amino acids

log(20) provides an upper bound on the entropy, and is thus a useful quantity to compare to. However, we agree with the reviewer that the entropy of amino acids according to their distribution in the overall proteome is an additional useful comparison point. We have added its value in the main text (4.2 bits).

– Page 6, left: "there is no evidence that selection at any step involves complex sequence motifs, or amino acid interactions, which would not be captured by the linear model" – It is not clear why the presence of such "motifs" would be expected in the first place, but there could still be strong selection against TCRs that recognize peptides in certain (ubiquitous) proteins, and this would not necessarily translate to simple "motifs" that are readily detectable

We agree and have included an additional discussion of the limitations of what type and magnitude of selection might be expected stressing this point.

– idem "validating the hypothesis of linear selection factors" – I am not sure if this is meant to imply that such linear selection factors truly drive selection. It would seem more likely that the observed differences are population-level consequences of a selection process that operates in a (stochastic) sequence-specific manner

We agree and have clarified this part of the text.

– Discussion "Consistent with these previous results, we can project our model parameters to build a single hydrophobicity index, which we observe to be significantly increased in positively selected cells (DP pos) versus DP pre, and decreased in single positive sets (Figure S21)." – this raises the question whether the single hydrophobicity index might already be sufficient to explain the differences seen in the repertoire

We have included an additional paragraph discussing the hydrophobicity score. We show that it varies between populations, in a way that is overall reflective of the selection process at work. However, if we use a classifier based only on the hydrophobicity score, it performs worse than 1gram classifiers (*Figure 4—figure supplement 14*).

– "This number is in agreement with our observations (Figure 5B) that ∼ 30 TCRs increases the AUC of correctly classifying these cells as CD4^+^ spleen vs CD4 apo from ∼ 0.59 to ∼ 0.87." – I think this is not a valid interpretation. First, as noted above, this phenomenon would be observed for any such classification. Second, there is nothing special to an AUC value of 0.87

We agree and have completely rewritten this part of the text to clarify that we are not suggesting our data validates a quorum sensing model.

– Methods: The Adam optimizer was used for the linear classifiers. Was this necessary (e.g., were the models underspecified) or could a deterministic optimizer have been used?

We have removed this mention, as an Adam optimizer was not necessary.

[Editors' note: further revisions were suggested prior to acceptance, as described below.]

Reviewer #3 (Recommendations for the authors):The authors have made many changes to the manuscript and added additional findings as well. I think the manuscript has been improved in many places. However, I suspect the revision is not complete. My main criticism had been the interpretation of imperfect classification as evidence for a quorum sensing mechanism. The authors state in their rebuttal letter that they agree with me and that they "radically rewrote" the corresponding text. However, the relevant paragraph in the discussion, the one starting with "Overall, the statistical population-level differences between populations of thymocytes and mature T cells.…", is entirely unchanged from the previous version. There have been some additional explanations inserted in the Results section, but these are not really correct (i.e. it is said that T cells are "weak learners", but the "learner" in this context is the classification algorithm built by the authors, not the T cell, which is the instance being classified.) So the same interpretation mistakes are unfortunately still present in the revised version. Perhaps the authors overlooked something.

We thank the reviewer for insisting on this point. We were too sloppy in our previous revision and the comment made us realize that we need to be more rigorous when making the link between leaky thymic selection and quorum sensing.

We have now removed the sentence about TCR being weak learners. In general, we have tried to refrain from interpreting our results in terms of collective decision making in the Result section, deferring our argument about quorum sensing to the Discussion.

We have extensively rewritten the relevant paragraph in the discussion, with the aim of better explaining the fundamental difference between the two discrimination tasks (self-reactive vs non self-reactive TCR in thymus on the one hand, and self vs foreign peptides discrimination by T cells in the periphery on the other). To tie the two together, we have developed a simple mathematical argument in the Methods following the modeling approach of Butler et al. 2013. However, we tried to remain careful about direct quantitative interpretations of our results in terms of number of cells and making clear that the two tasks are different.